# Mechanical force of uterine occupation enables large vesicle extrusion from proteostressed maternal neurons

Guoqiang Wang[1†], Ryan J Guasp[1†], Sangeena Salam[1†], Edward Chuang[1], Andrés Morera[1], Anna J Smart[1], David Jimenez[1], Sahana Shekhar[1], Emily Friedman[1], Ilija Melentijevic[1], Ken C Nguyen[2], David H Hall[2], Barth D Grant[1], Monica Driscoll[1]*

[1]Department of Molecular Biology and Biochemistry, Nelson Biological Laboratories, Rutgers, The State University of New Jersey, Piscataway, United States; [2]Department of Neuroscience, Albert Einstein College of Medicine, Bronx, United States

*For correspondence: driscoll@dls.rutgers.edu

†These authors contributed equally to this work

Competing interest: The authors declare that no competing interests exist.

**Abstract** Large vesicle extrusion from neurons may contribute to spreading pathogenic protein aggregates and promoting inflammatory responses, two mechanisms leading to neurodegenerative disease. Factors that regulate the extrusion of large vesicles, such as exophers produced by proteostressed *C. elegans* touch neurons, are poorly understood. Here, we document that mechanical force can significantly potentiate exopher extrusion from proteostressed neurons. Exopher production from the *C. elegans* ALMR neuron peaks at adult day 2 or 3, coinciding with the *C. elegans* reproductive peak. Genetic disruption of *C. elegans* germline, sperm, oocytes, or egg/early embryo production can strongly suppress exopher extrusion from the ALMR neurons during the peak period. Conversely, restoring egg production at the late reproductive phase through mating with males or inducing egg retention via genetic interventions that block egg-laying can strongly increase ALMR exopher production. Overall, genetic interventions that promote ALMR exopher production are associated with expanded uterus lengths and genetic interventions that suppress ALMR exopher production are associated with shorter uterus lengths. In addition to the impact of fertilized eggs, ALMR exopher production can be enhanced by filling the uterus with oocytes, dead eggs, or even fluid, supporting that distention consequences, rather than the presence of fertilized eggs, constitute the exopher-inducing stimulus. We conclude that the mechanical force of uterine occupation potentiates exopher extrusion from proximal proteostressed maternal neurons. Our observations draw attention to the potential importance of mechanical signaling in extracellular vesicle production and in aggregate spreading mechanisms, making a case for enhanced attention to mechanobiology in neurodegenerative disease.

## eLife assessment

This **important** study explores the potential influence of physiologically relevant mechanical forces on the extrusion of vesicles from *C. elegans* neurons. The authors provide **compelling** evidence to support the idea that uterine distension per se can induce vesicular extrusion from adjacent neurons. Overall, this work will be of interest to neuroscientists and investigators in the extracellular vesicle and proteostasis fields.

**eLife digest** Neurons are specialized cells in the brain and nervous system that transmit signals between the brain and the rest of the body, enabling humans and animals to react to internal and external stimuli. For this communication system to function effectively, neurons must remain healthy.

Neurons maintain their function in a variety of ways, including by removing excess or damaged cellular components (such as organelles and protein aggregates) that could compromise neuron function. One way to do this is by extruding organelles and aggregates. During 'extrusion events', the material to be removed is gathered within a budding portion of the plasma membrane, which forms a vesicle that ejects the material from the neuron. However, the factors driving the extrusion process remained unknown.

To investigate, Wang, Guasp, Salam et al. conducted experiments in the roundworm *Caenorhabditis elegans,* finding that the number of extrusion events in a certain type of neuron increases at the peak of reproduction. More specifically, a greater number of extrusion events were associated with the presence of fertilized eggs, which accumulate in the uterus before they are laid. Disrupting eggs, sperm or the fertilization process suppressed the increase in extrusion events, suggesting the presence of fertilized eggs is responsible.

To determine how the eggs might trigger extrusion events, Wang et al. stretched the uterus using dead eggs, unfertilized eggs or by injecting fluid, finding that each of these approaches increased the number of extrusion events. Further analysis suggests that this mechanical stretching of the uterus signals to the neurons that reproduction has started, encouraging the neurons to remove old components and optimize their function. Wang et al. hypothesize that this stretch response could support neuronal behaviors that aid in successful reproduction, such as sensing food and selecting where to lay eggs.

The findings increase our understanding of the factors that trigger vesicle extrusion in living organisms. These observations could have implications for human neurodegenerative diseases such as Alzheimer's disease, in which protein aggregates accumulate in neurons. It is possible that mechanical signals generated by factors associated with Alzheimer's disease, such as high blood pressure, could influence neuronal extrusion and contribute to some of the mechanisms underlying aggregate transfer in neurodegenerative diseases.

## Introduction

In neurodegenerative disease, prions and protein aggregates can transfer among cells of the nervous system to promote pathology spread (*Peng et al., 2020*; *Davis et al., 2018*). Determination of the factors that enhance or deter pathological transfer is, therefore, a central goal in the effort to clinically address neurodegenerative disease. Study of aggregate transfer in the context of the mammalian brain is a major experimental challenge as events are rare, sporadic, and transiently apparent, and tissue is not easily accessible for in vivo observation. We model aggregate transfer by proteostressed ALMR touch receptor neurons in the living *C. elegans* nervous system (*Melentijevic et al., 2017*; *Cooper et al., 2021*; *Arnold et al., 2023*), an experimental system that enables molecular and genetic manipulation and evaluation in a physiological context, directly through the transparent cuticle (*Corsi et al., 2015*).

More specifically, *C. elegans* adult neurons can extrude large vesicles called exophers (~5 μm; 100 X larger than exosomes) that carry potentially deleterious proteins and organelles out of the neuron (*Melentijevic et al., 2017*; *Cooper et al., 2021*; *Arnold et al., 2023*). Disrupting proteostasis via diminished chaperone expression, autophagy, or proteasome activity, or over-expressing aggregating proteins like human Alzheimer's disease associated fragment $A\beta_{1-42}$, expanded polyglutamine Q128 protein, or high concentration mCherry fluorophore, increases exopher production from the affected neurons. Neurons that express proteotoxic transgenes maintain higher functionality if those neurons produce exophers as compared to those that do not, suggesting that exopher-genesis can be neuroprotective, at least in young adult neurons. Extruded exopher contents can be transferred to neighboring glial-like hypodermal cells for content degradation in the lysosomal network (*Wang et al., 2023*). Several mammalian models feature similar biology (*Nicolás-Ávila et al., 2020*; *Lampinen et al., 2022*; *Davis et al., 2014*; *Nicolás-Ávila et al., 2021*; *Nicolás-Ávila et al., 2022*), and thus

we speculate that the basic transfer biology represents a conserved process that can be recruited for animal-wide proteostasis balance.

Neuronal exophers are generated only in adult animals, with an unexpected pattern of a peak at young adult days 2–3 and then later in age with more variable onset (*Melentijevic et al., 2017*). While using chemical regent 5-fluoro-2'-deoxyuridine (FUdR) to block progeny production for aging studies, we found that blocking reproduction strongly limited the early adult peak of exopher production. Here, we report data that support that early adult exopher production is sensitive to the load of eggs in the reproductive tract. We document that uterine expansion, rather than chemical signals emanating from fertilized eggs, correlates strongly with the level of exopher production and suggest a model in which mechanical signaling, normally induced across generations from egg to parent via uterine stretch, is a license for proteostressed neurons to release potentially toxic materials in large extracellular vesicles.

Mechanical signaling exerts a profound impact on virtually all cell types, and has been implicated in traumatic brain injury and neurodegenerative disease, yet remains poorly understood (*Hall et al., 2021*). Our observations direct enhanced experimental attention to studies on how mechanical force can influence extracellular vesicle formation and aggregate transfer in the living brain and in neurodegenerative disease.

## Results

The six *C. elegans* gentle touch receptor neurons (AVM, ALML, and ALMR located in the anterior body, and PVM, PLML and PLMR located in the posterior body) can be readily visualized in vivo by expression of fluorescent proteins under the control of the touch neuron-specific *mec-4* promoter (*Figure 1A*). We commonly monitor exophers extruded by the ALMR neuron, which typically produces more exophers than the other touch neurons (*Melentijevic et al., 2017*; *Arnold et al., 2020*), using a strain in which fluorophore mCherry is highly expressed in touch neurons and is avidly eliminated (strain ZB4065 *bzIs166*[P$_{mec-4}$::mCherry], hereafter referred to as mCherryAg2 for simplicity). ALMR exopher production occurs with a distinctive temporal profile, such that at the L4 larval stage ALMR rarely, if ever, produces an exopher, but in early adult life, the frequency of exopher events increases, typically reaching a peak of 5–20% of ALMR neurons scored at adult day 2 (Ad2); exopher detection then falls to a low baseline level after Ad3 (*Melentijevic et al., 2017*; *Arnold et al., 2020*), a pattern that parallels adult reproduction (*Figure 1B*). Late in life, exophers can reappear with variable frequency, but here we focus on the young adult exopher generation.

### Sterility-inducing drug FUdR suppresses ALMR exophergenesis

In experiments originally designed to study exopher events in aging animals, we sought to generate age-synchronized populations by blocking progeny production from early adult stages using DNA synthesis inhibitor FUdR. Unexpectedly, we found that 51 µM FUdR strongly suppressed exopher events as quantitated at Ad2 (*Figure 1C*). FUdR is commonly used in *C. elegans* to inhibit the proliferation of germline stem cells and developing embryos, but has also been noted to impair RNA metabolism (*Burnaevskiy et al., 2018*) and improve adult proteostasis (*Angeli et al., 2013*). To probe which FUdR outcome might confer exopher suppression, we first addressed whether disruption of progeny production by alternative genetic means might suppress exophergenesis, which would implicate a viable germline as a factor in exopher modulation.

### Germline elimination and germline tumors suppress ALMR exopher production

In the assembly line-like hermaphrodite *C. elegans* gonad, germline stem cells close to the signaling distal tip cell (DTC) proliferate by mitosis (diagram of *C. elegans* gonad and germ cell development in *Figure 1D*). We disrupted germ cell production using the *glp-4(bn2ts)* valyl-tRNA synthetase 1 mutant. At the restrictive temperature (25 °C), *glp-4(bn2ts)* causes germ cell arrest during the initial mitotic germ cell divisions, effectively eliminating the germline (*Beanan and Strome, 1992*). We constructed an mCherryAg2; *glp-4(bn2ts)* strain and quantitated ALMR exopher production in animals grown at the restrictive temperature, scoring exophers during early adulthood (*Figure 1E*). ALMR neurons in

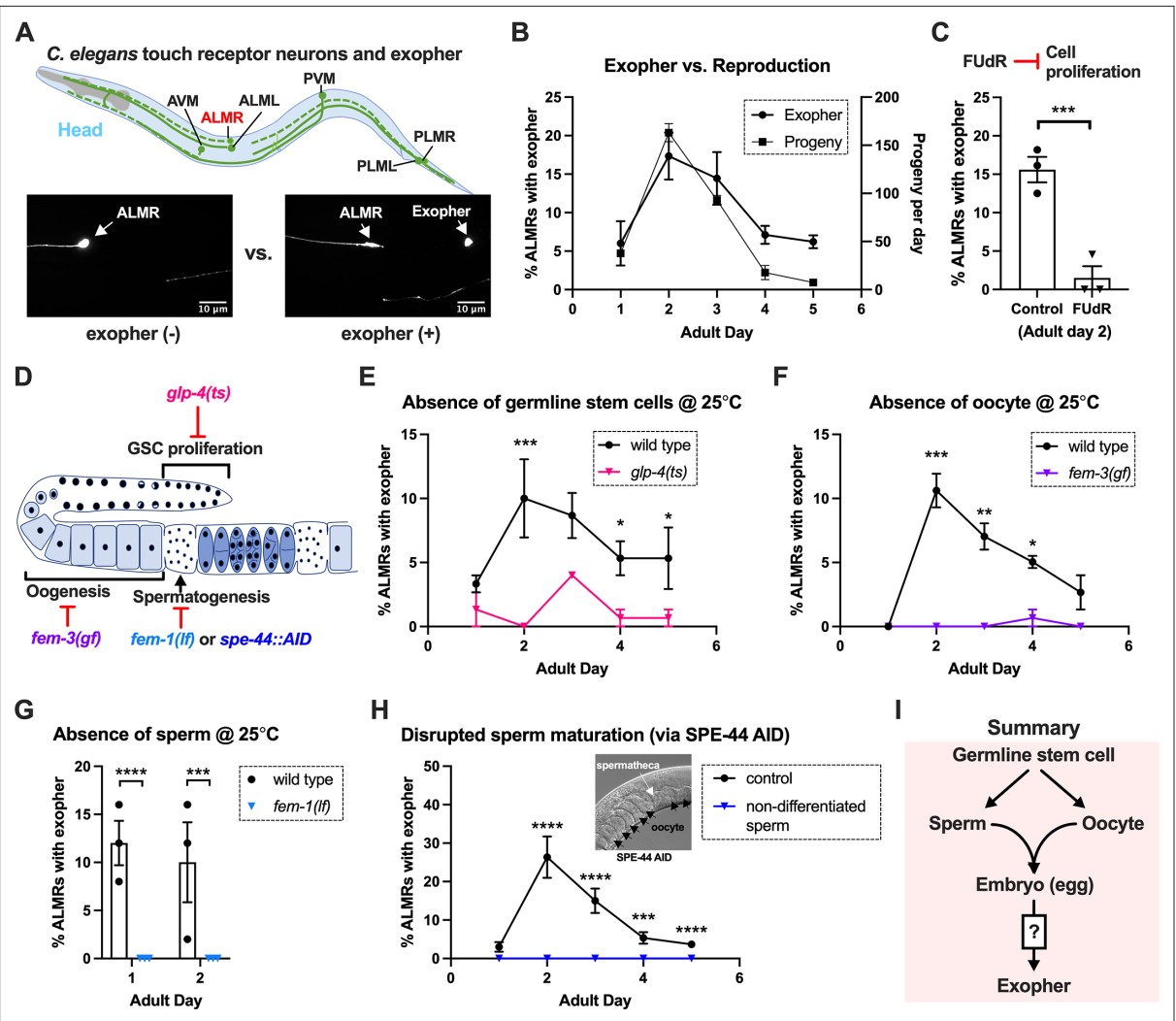

**Figure 1.** Exophergenesis is dependent on the presence of the germline, ooyctes, and sperm. (**A**) Exophers produced by ALMR are readily visualized in living *C. elegans*. Top, positions of the six *C. elegans* touch receptor neurons: AVM (Anterior Ventral Microtubule cell), ALMR (Anterior Lateral Microtubule cell, Right), ALML (Anterior Lateral Microtubule cell, Left), PVM (Posterior Ventral Microtubule cell, Right), PLMR (Posterior Lateral Microtubule cell, Right), and PLML (Posterior Lateral Microtubule cell, Left). Bottom panels are representative pictures (n > 100, scale bar = 10 μm) of an ALMR neuron without (lower left) or with (lower right) exopher production from strain ZB4065 *bzIs166*[P*mec-4*::mCherry], which expresses elevated mCherry in the touch receptor neurons. Over-expression of mCherry in *bzIs166* is associated with enlargement of lysosomes and formation of large mCherry foci that often correspond to LAMP::GFP-positive structures; ultrastructure studies reveal considerable organelle morphological change not seen in low reporter-expression neurons; polyQ74, polyQ128, Aβ$_{1-42}$ over-expression all increase exophers (***Melentijevic et al., 2017***; ***Arnold et al., 2023***). Most genetic compromise of different proteostasis branches--heat shock chaperones, proteasome, and autophagy--enhance exophergenesis, supporting exophergenesis as a response to proteostress. (**B**) Both exopher production and reproduction typically peak around Ad2 in the P*mec-4*::mCherry strain. Left axis: the frequency of exopher events in the adult hermaphrodite *C. elegans* strain ZB4065 at 20 °C; Mean ± SEM of nine independent trials (~50 animals in each trial). Right axis: daily progeny count (Mean ± SEM) from 10 wild-type N2 hermaphrodites. (**C**) 5-fluoro-2'-deoxyuridine (FUdR), which inhibits progeny production, suppresses early adult exopher production. Data are the percentage of ALMR exopher events among >50 Ad2 hermaphrodites in each trial (total of 3 independent trials) for strain ZB4065 at 20 °C in absence (control) or presence of 51 μM FUDR. ***p<0.001 in Cochran–Mantel–Haenszel test. (**D**) Illustration of the roles of germline development genes tested for impact on exophers. The *C. elegans* reproductive system comprises a bilobed gonad in which germ cells (light blue, dark nuclei) develop into mature oocytes, which are fertilized in the spermatheca (sperm indicated as dark dots) and held in the uterus until about the 30-cell gastrulation stage, at which point eggs (dark blue) are laid. Indicated are the steps impaired by germline developmental mutations we tested. (**E**) Germline stem cells are required for efficient exopher production. Data show the percentage of ALMR exopher events (Mean ± SEM) among 50 adult hermaphrodite *C. elegans* that express the wild-type GLP-4 protein or the GLP-4(ts) protein encoded by *glp-4(bn2)* at 25 °C. All animals express P*mec-4*mCherry, and the *glp-4(ts)* mutants lack a germline when reared at the restrictive temperature (25 °C). Eggs collected from both wild-type and the *glp-4(ts)* mutants were grown at 15 °C for 24 hr before being shifted to 25 °C (at L1 stage) to enable development under restrictive conditions; three independent trials of 50 animals represented by each dot; ***p<0.001 or *p<0.05 in Cochran–Mantel–Haenszel test. Note that in wild-type (WT), temperature shift normally induces a modest elevation in exopher numbers (typically a

*Figure 1 continued on next page*

*Figure 1 continued*

few % increase, supplemental data in **Cooper et al., 2021**) and is thus not itself a factor in exopher production levels. (**F**) Oogenesis is required for efficient exopher production. Spermatogenesis occurs but oogenesis is blocked when *fem-3(gf)* mutant hermaphrodites are cultured at 25°C. Data show the percentage of ALMR exopher events (Mean ± SEM) in hermaphrodite *C. elegans* that express the wild-type FEM-3 protein or the temperature-dependent gain-of-function (*gf*) FEM-3 protein (25 °C; three independent trials, n=50/trial), ***$p<0.001$; **$p<0.01$; or *$p<0.05$ in the Cochran–Mantel–Haenszel test. (**G**) Spermatogenesis is required for efficient exopher production. There is no spermatogenesis in *fem-1(lf)* at the restrictive temperature of 25 °C, while oogenesis is normal in the hermaphrodite. Shown is the percentage of ALMR exopher events (Mean ± SEM) in adult hermaphrodites that express the wild-type FEM-1 protein or the temperature-dependent loss of function (*lf*) FEM-1 protein (25 °C; three independent trials, 50 animals/trial). ****$p<0.0001$ or ***$p<0.001$ in Cochran–Mantel–Haenszel test. (**H**) Spermatogenesis is required for efficient exopher production. Data show the percentage of ALMR exopher events (Mean ± SEM) in hermaphrodite *C. elegans* that express the SPE-44::degron fusion. SPE-44 is a critical transcription factor for spermatogenesis (**Kulkarni et al., 2012**), and is tagged with a degron sequence that enables targeted degradation in the presence of auxin in line ZB4749. In the auxin-inducible degron (AID) system, auxin is added to the plates in 0.25% ethanol, so 'control' is treated with 0.25% ethanol and 'no sperm' is treated with 1 mM auxin applied to plates in 0.25% ethanol from egg to adult day 1; 4–6 independent trials of 50 animals per trial. ****$p<0.0001$ or ***$p<0.001$ in Cochran–Mantel–Haenszel test. Note that under no sperm or non-functional sperm production, oocytes still transit through the spermatheca and enter the uterus (as shown by the DIC picture); unfertilized oocytes can be laid if the egg-laying apparatus is intact. (**I**) Summary: Genetic interventions that block major steps of germ cell development strongly block ALMR exophergenesis in the adult hermaphrodite. The dual requirement for sperm and oocytes suggests that fertilization and embryogenesis are required events for inducing ALMR exophergenesis.

The online version of this article includes the following source data for figure 1:

**Source data 1.** Daily progeny count for panel B, and exopher score for panels B, C, E, F, G, H.

the germline-less *glp-4* mutant produced significantly fewer exophers on Ad1-4 as compared to age- and temperature-matched controls.

## Both oogenesis and spermatogenesis are critical for early adult exopher production

Given that lack of functional germ cells impaired neuronal exopher production, we sought to test whether oocytes or sperm might be specifically required for exophergenesis, taking advantage of *C. elegans* genetic reagents available for the manipulation of gamete development.

### Oogenesis is required for the peak of early adult exophergenesis

In the *C. elegans* hermaphrodite, the differentiation of germline stem cells begins during the L4 larval stage with spermatogenesis, after which sperm production is shut down and oogenesis begins (**Zanetti et al., 2012**). Genetic mutants that produce only sperm or only oocytes have been well characterized. To test a mutant that produces sperm but no oocytes, we employed *fem-3(q20ts)*, a temperature-sensitive gain-of-function mutant that causes germ cells to exclusively differentiate into sperm (**Ellis and Schedl, 2007**; **Ahringer and Kimble, 1991**). We constructed an mCherryAg2; *fem-3(q20ts)* line, reared animals at non-permissive temperature 25 °C, and scored ALMR exophers in mutant and control animals during early adulthood. In the sperm-only, oocyte-deficient *fem-3(q20ts)* background, exophergenesis is mostly diminished over the first five days of adulthood (*Figure 1F*). We infer that oocytes must be present for early adult ALMR exopher production and conclude that sperm alone are not sufficient to drive exopher elevation.

### Spermatogenesis is required for the peak of early adult exophergenesis

The presence of functional sperm can stimulate ovulation to trigger oocyte maturation. To assess neuronal exopher production in the reciprocal reproductive configuration in which animals have oocytes but no sperm, we disrupted spermatogenesis using two approaches. First, we used a temperature-sensitive *fem-1* mutation to produce animals with oocytes but no sperm (**Doniach and Hodgkin, 1984**). We crossed *fem-1(fc17ts)* into the mCherryAg2 strain and examined exopher production from ALMR neurons in hermaphrodites at the restrictive temperature of 25 °C. We found that ALMR exopher production in *fem-1(fc17)* mutants was suppressed at 25 °C (*Figure 1G*).

Second, we used the auxin-inducible degron system (AID) (**Nishimura et al., 2009**; **Zhang et al., 2015**) to degrade SPE-44, an essential protein required for spermatid differentiation (**Kasimatis et al., 2018**; **Kulkarni et al., 2012**). In the AID system, the addition of auxin to the culture induces rapid degradation of proteins genetically tagged with a specific auxin-dependent degron sequence. AID targeting of SPE-44-degron is highly effective in disrupting sperm maturation (**Kasimatis et al., 2018**).

We treated strain ZB4749 (fxIs1[P*pie-1*::TIR1::mRuby] zdIs5[P*mec-4*::GFP]; bzIs166[P*mec-4*::mCherry]; *spe-44*(fx110[*spe-44*::degron])) with auxin during larval developmental stages to block sperm maturation and then transferred Ad1 animals to NGM plates without auxin, a condition that disrupted sperm maturation but allowed oocyte generation. We find that blocking sperm maturation by targeting SPE-44 for degradation abolished ALMR exopher production (*Figure 1H*) even though oogenesis is not significantly impacted (*Kasimatis et al., 2018*). We infer that functional sperm must be present for early adult ALMR exopher production and conclude that oogenesis alone is not sufficient to drive exopher elevation in early adult life.

## Fertilization and early embryonic divisions are required for the early adult exophergenesis peak

The requirement of sperm and oocytes for neuronal exopher production raises the obvious question as to whether fertilized eggs/embryos are required for ALMR exophergenesis (*Figure 1I*). *C. elegans* genes impacting fertilization and embryonic development have been studied in exquisite detail (*Greenstein, 2005*; *Stein and Golden, 2018*; *Rose and Gönczy, 2014*; *Schneider and Bowerman, 2003*). We tested embryonic development genes known for roles at particular stages for impact on neuronal exopher levels using RNAi knockdown approaches.

In *C. elegans* fertilization, as the mature oocyte encounters the sperm-filled spermatheca, a single sperm enters the mature ovulated oocyte (fertilization time 0), triggering the rapid events of egg activation, which include polyspermy block, eggshell formation, completion of meiotic divisions and extrusion of polar bodies. The multi-step eggshell formation (*Figure 2A*) is completed within 5 min of sperm entry, by which time the zygote has been passed into the uterus, where the maternal chromosomes execute both meiotic divisions (meiosis accomplished by ~20 min after fertilization). After the establishment of egg polarity cues, the first mitotic cell division occurs ~40 min after fertilization (*Stein and Golden, 2018*; *Schneider and Bowerman, 2003*). Eggs are held in the mother's uterus until they are laid at the ~30- cell stage (gastrulation). In wild-type, ~8 fertilized eggs occupy the uterus when egg laying begins at ~6 hr of adult life (*Medrano and Collins, 2023*).

Our data revealed an unexpected theme: we found that disruption of tested early-acting genes essential for eggshell assembly (chitin-binding domain protein *cbd-1 González et al., 2018*; *Figure 2B*, chitin precursor synthesis gene *gna-2 Johnston et al., 2006*; *Figure 2C and D*, permeability barrier-required CDP-ascarylose synthesis gene *perm-1 Olson et al., 2012*; *Figure 2E*), and/or needed for the progression through the 1 cell or 2 cell stages of embryonic development (*Severson et al., 2002*; *Rappleye et al., 1999*; *Rappleye et al., 2003*; *Benenati et al., 2009*) (profilin *pfn-1* and coronin-related *pod-1* (*Figure 2F*)), cause potent suppression of ALMR exophergenesis (*Figure 2G*). In contrast, RNAi knockdowns of genes playing important roles for later stage embryonic development (4-cell to 8- cell stages *mex-3*, *mom-2*, gastrulation genes *end-1/–3* or *gad-1*) (*Figure 2F*) confer negligible impact on ALMR exophergenesis (*Figure 2H*). Our data support a fertilization requirement for neuronal exopher stimulation and suggest that the exophergenesis signal/condition that promotes the early adult wave of exophergenesis is associated with the very earliest stages of embryonic development. Of note, disruption of eggshell biosynthesis has the immediate downstream consequence of disruption of early polarity establishment and first divisions, such that genetic separation of eggshell production from first division proficiency is not possible. For example, *pod-1* RNAi has been reported to be associated with eggshell deficits (*Rappleye et al., 1999*), although no eggshell deficits are oberseved for *pfn-1* RNAi (*Sönnichsen et al., 2005*). We thus conclude that either eggshell integrity or biochemical events associated with the first embryonic cell divisions are required for neuronal exophergenesis.

## Restoring fertilized eggs later in life can extend ALMR exophergenesis

Having found that fertilized eggs are necessary for the early adult elevation in exopher production, we asked whether the presence of eggs is sufficient for stimulation of exopher production by introducing fertilized eggs later in adult life when they are not normally present. In *C. elegans*, sperm are made in the L4 stage and are stored in the spermatheca to fertilize oocytes that mature in the adult. Sperm are limiting for hermaphrodite self-fertilization, with more oocytes made than can be fertilized by self-derived sperm. Unmated hermaphrodites thus cease egg laying around adult day 4 because they run out of self-supplied sperm. However, if males mate with hermaphrodites, increased progeny

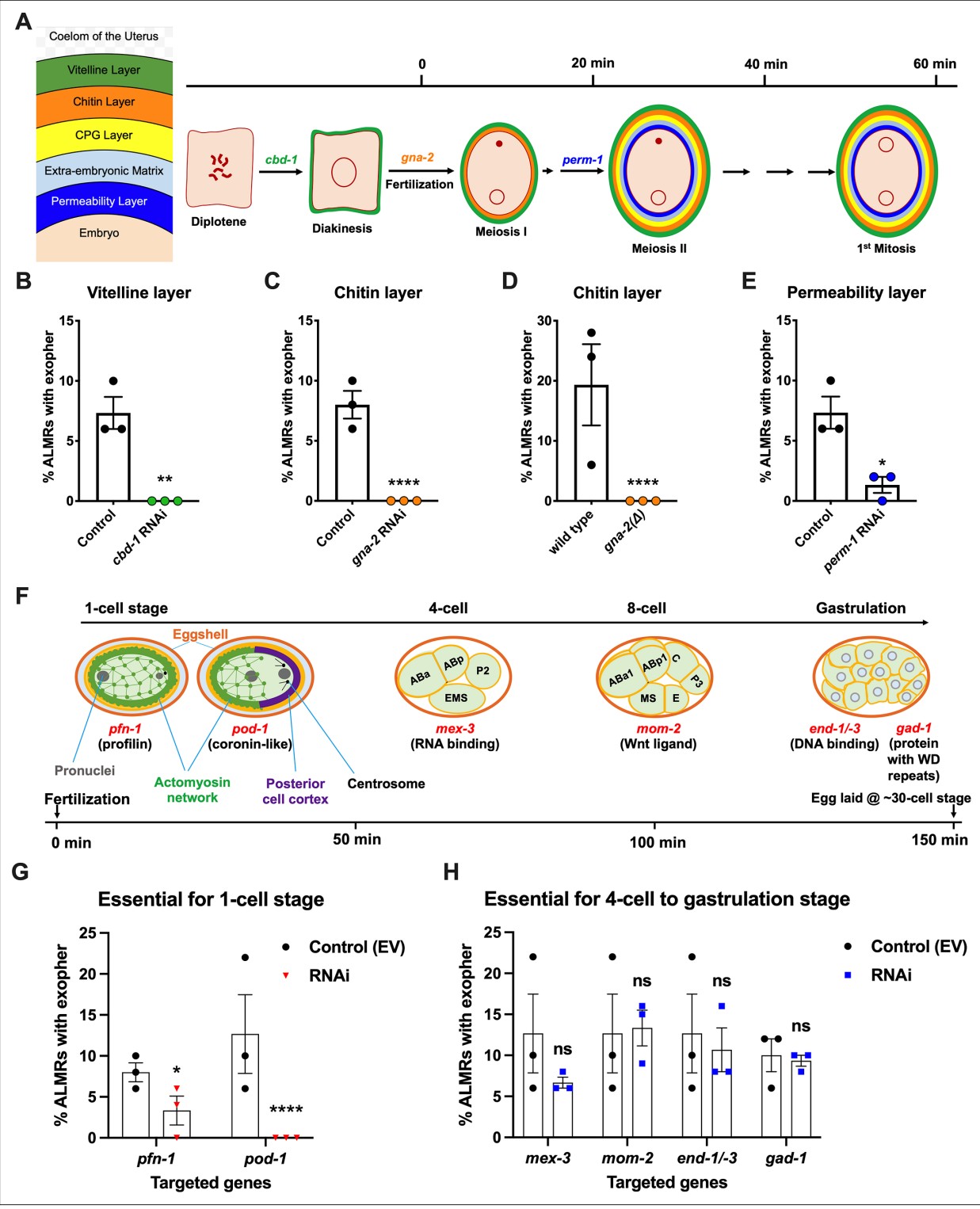

**Figure 2.** Events required for adult day 2 (Ad2) elevation of exopher production occur during the earliest stages consequent to fertilization and are largely completed by the 4-cell stage. (**A**) Diagram of eggshell layers, post-fertilization timeline for layer formation, and indication of steps at which RNAi disrupts eggshell biogenesis. The formation of the multilayered eggshell is accomplished via a hierarchical assembly process from outside to inside, with outer layers required for later formation of inner layers. The outmost vitelline layer assembles in part prior to fertilization, dependent on chitin-binding domain protein (CBD-1) *González et al., 2018*. The next eggshell layer is made up of chitin, which confers eggshell stiffness and requires the *gna-2*-encoded enzyme glucosamine-6-phosphate N-acetyltransferase (GNPNAT1) for precursor biosynthesis (*Johnston et al., 2006*). The innermost lipid layer of eggshell is called the permeability layer, which is lipid-rich and is needed to maintain osmotic integrity of the embryo. PERM-1

*Figure 2 continued on next page*

*Figure 2 continued*

(*Olson et al., 2012*) (among others, including FASN-1 [*Rappleye et al., 2003*], POD-1 [*Rappleye et al., 1999*], and EMB-8 [*Benenati et al., 2009*]) is required for permeability barrier formation (*Stein and Golden, 2018*; *Johnston and Dennis, 2012*). Note that eggshell biogenesis is critical for polyspermy barrier, spermathecal exit, meiotic chromosome segregation, polar body extrusion, AP polarization and internalization of membrane and cytoplasmic proteins, and correct first cell divisions (*Johnston and Dennis, 2012*), so genetic separation of eggshell malformation from the earliest embryonic formation is not possible. (**B**) *cbd-1*, a gene encoding an essential component of the eggshell vitelline layer, is critical for Ad2 exopher elevation. Exopher scores in Ad2 animals (strain ZB4757: bzIs166[P$_{mec-4}$::mCherry] II) that were treated with RNAi against *cbd-1*, total of three trials (50 hermaphrodites per trial), \*\**p*<0.01 in Cochran–Mantel–Haenszel test, as compared to the empty vector (EV) control. (**C**) *gna-2*, a gene required for chitin precursor biosynthesis and chitin layer formation, is critical for Ad2 exopher elevation. (**C**) Exopher scores in Ad2 animals treated from the L1 stage with RNAi against *gna-2*. The *gna-2* gene encodes enzyme glucosamine-6-phosphate N-acetyltransferase (GNPNAT1) required for chitin precursor biosynthesis. (**D**) Exopher scores in Ad2 animals harboring a null mutation in the essential *gna-2* gene. *gna(Δ)* homozygous null worms are GFP negative progeny of stain ZB4941: bzIs166[P$_{mec-4}$::mCherry]; *gna-2(gk308)* I/hT2 [*bli-4(e937) let-?(q782)*] qIs48[P$_{myo-2}$::GFP; P$_{pes-10}$::GFP; P$_{ges-1}$::GFP] (I;III). Data represent a total of three trials (50 hermaphrodites per trial), \*\*\*\* *p*<0.0001 in Cochran–Mantel–Haenszel test, as compared to wild-type animals. (**E**) *perm-1*, a gene required for permeability barrier synthesis, is critical for Ad2 exopher elevation. Exopher scores in Ad2 animals treated with RNAi against *perm-1*, which encodes a sugar modification enzyme that acts in the synthesis of CDP-ascarylose. Data represent a total of 3 trials (50 hermaphrodites per trial), \*p<0.05 in Cochran–Mantel–Haenszel test, as compared to the EV control. Note that previously characterized strong exopher suppressors *pod-1*, *emb-8*, and *fasn-1* (*Melentijevic et al., 2017*; *Cooper et al., 2021*) are also needed for egg shell permeability barrier layer formation (*Rappleye et al., 1999*; *Benenati et al., 2009*). (**F**) Diagram of select genes required for specific stages of embryonic development. *pfn-1* RNAi (arrest at the one-cell stage *Schonegg et al., 2014*; *pod-1* RNAi (arrest at the two-cell stage *Luke-Glaser et al., 2005*; arrest stage phenotype for other genes is not precisely documented, but these genes play significant roles at the indicated stages; images annotated according to WormAtlas (https://doi.org/10.390/wormatlas.4.1). (**G**) RNAi targeting of early acting embryonic development genes lowers exopher production. Exopher scores from Ad2 animals that were treated with RNAi against genes characterized to be essential for 1-cell to 2-cell stage embryonic development. Total of three trials (50 hermaphrodites per trial). \*p<0.05 or \*\*\*\*p<0.0001 in Cochran–Mantel–Haenszel test, as compared to the empty vector control. Strong exopher suppressor *pod-1* has been previously reported (*Melentijevic et al., 2017*). We found RNAi directed against gene *emb-8* (as early as 2 cell arrest, but arrest at the 1- to 50 cell stage reported [*Schierenberg et al., 1980*]) to be more variable in outcome (not shown). (**H**) RNAi targeting of genes that disrupt 4 cell stage and later embryonic development does not alter exopher production levels. Exopher scores in Ad2 animals that were treated with RNAi against genes that are essential for 4 cell to gastrulation stages of embryonic development. Total of three trials (50 hermaphrodites per trial). ns, not significant in Cochran–Mantel–Haenszel test, as compared to the empty vector control.

The online version of this article includes the following source data for figure 2:

**Source data 1.** Exopher score for panels B, C, D, E, G, H.

numbers can be produced due to increased sperm availability. More germane to our experiment, if males are mated to the reproductively senescing hermaphrodite to replenish sperm, hermaphrodites can produce fertilized eggs for a few additional days (*Figure 3A*).

To determine if the presence of eggs might be sufficient to drive exopher production after Ad3, we mated males to Ad3 reproductively senescing hermaphrodites. We found that restored egg production is associated with increased and extended exopher production, provided that hermaphrodites were mated to fertilization-proficient males (*Figure 3B*). We conclude that adult ALMR exophergenesis is driven by the presence of fertilized eggs and that the older age decrease in exopher production (~Ad4) under standard culture conditions is more likely attributed to the lack of fertilized egg accumulation at this life stage, rather than to the existence of a chronological limit on the biochemical capacity to elevate exopher levels at older ages.

## Genetically induced egg retention elevates ALMR exopher production

Fertilized eggs might release a signal or create a condition that stimulates exophergenesis. If such an influence were limiting, increasing the egg concentration in the body might increase exopher numbers. To address whether elevating the young adult egg load can increase exophers, we took multiple distinctly-acting genetic approaches to limit egg laying and promote egg retention in the body (*Figure 3C*). We tested four genetic conditions that lower or block egg-laying: a null mutation of prolyl hydroxylase *egl-9*, for which disruption induces a mild egg-laying defect (*Trent et al., 1983*); a null allele of proprotein convertase subtilisin/kexin type 2 *egl-3* (*Salem et al., 2018*) that perturbs neuropeptide processing and confers a severe egg-laying defective phenotype (*Trent et al., 1983*); a reduction-of-function mutation in SOX transcription factor SEM-2 (*sem-2(n1343)*) that eliminates production of the sex myoblasts needed to generate egg laying muscle; and RNAi directed against the *lin-39* homeobox transcription factor HOXA5 ortholog required for vulval cell fate specification (*Wagmaister et al., 2006*; *Sternberg, 2005*).

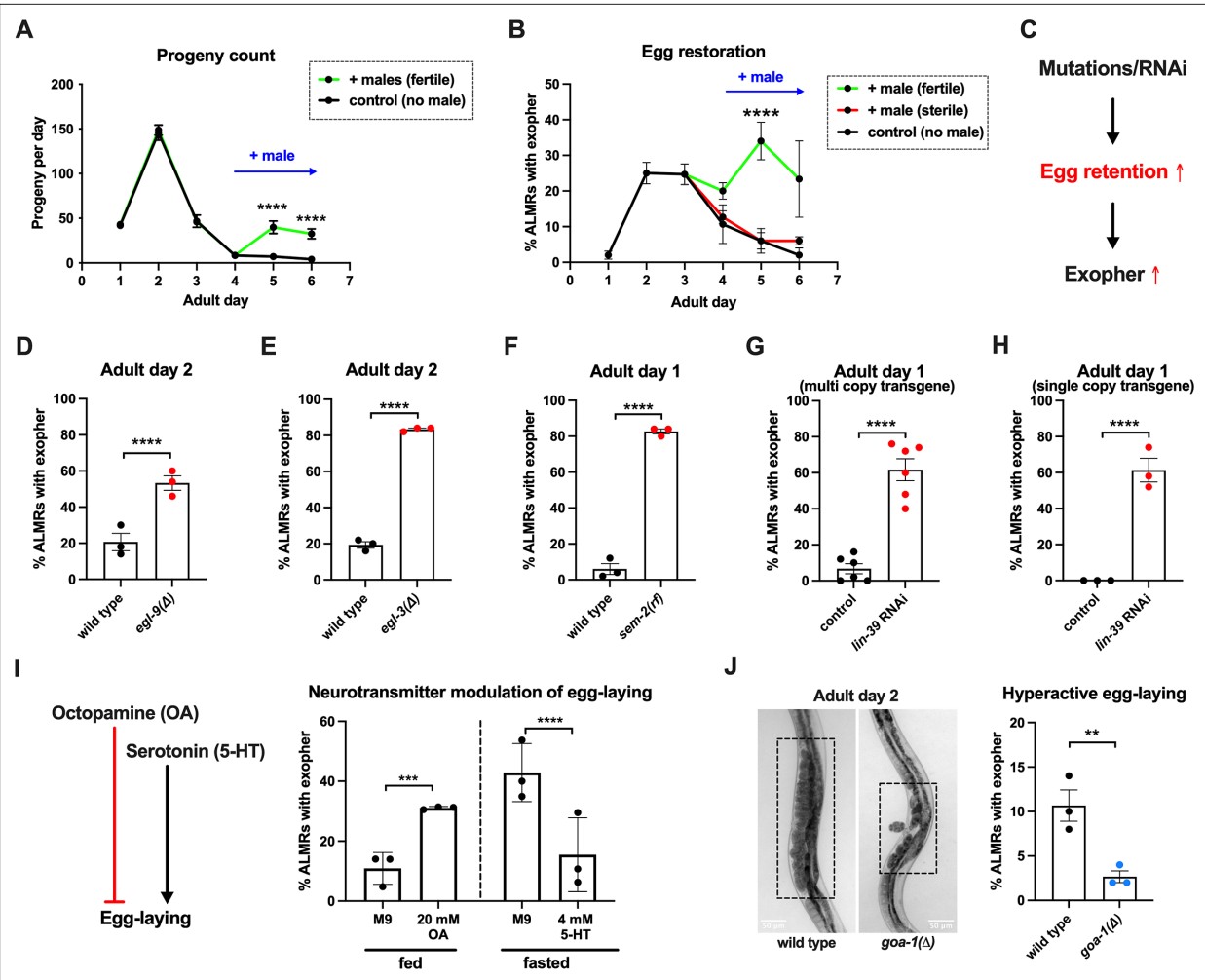

**Figure 3.** Anterior Lateral Microtubule cell (ALMR) exophergenesis levels are markedly influenced by the number of fertilized eggs retained in the uterus. (**A**) Later life mating extends the *C. elegans* reproductive period. Progeny count for each wild-type hermaphrodite in the presence (green) or absence (black) of males (1 hermaphrodite +/-5 males) from Ad1 to Ad6. Males are present from Ad4 to Ad6. Total of 12 wild-type hermaphrodites scored for each condition. Data shown are mean ± SEM. ****$p<0.0001$ in two-way ANOVA with Šídák's multiple comparisons test. (**B**) Introducing fertilized eggs in late adulthood extends the period of elevated exophergenesis. Males carrying functional (green) or nonfunctional (red) sperm (*spe-45(tm3715)*) were added to plates housing hermaphrodites as endogenous stores of sperm are depleted at Ad3. Data shown are mean ± SEM of percentage hermaphrodite ALMR neurons exhibiting an exopher event on days indicated. Total of three trials, and 50 hermaphrodites for each treatment at a single time point. ****$p<0.0001$ in Cochran–Mantel–Haenszel test, as compared to the sterile male group or no male control. (**C**) Hypothesis: Genetic interventions (mutations/RNAi) that increase egg retention increase ALMR exophergenesis. (**D**) Genetic interventions that induce egg retention elevate exopher levels. *egl-9(sa307)*. ALMR exopher scores Ad2. Strain ZB4757: bzIs166[P*mec-4*::mCherry] II vs. strain ZB4772: bzIs166[P*mec-4*::mCherry] II; *egl-9(sa307)* V. Total of three trials (50 worms per trial, each trial one dot); **** $p<0.0001$ in Cochran–Mantel–Haenszel test, as compared to the wild-type control. (**E**) Genetic interventions that induce egg retention elevate exopher levels. *egl-3(gk238)*. ALMR exopher scores Ad1. Strain ZB4757: bzIs166[P*mec-4*::mCherry] II vs. strain ZB4904: bzIs166[P*mec-4*::mCherry] II; *egl-3(gk238)* V. Total of three trials (50 worms per trial, each trial one dot); ****$p<0.0001$ in Cochran–Mantel–Haenszel test, as compared to the wild-type control. (**F**) Genetic interventions that induce egg retention elevate exopher levels. *sem-2(n1343)*. ALMR exopher scores Ad1. Strain ZB4757: bzIs166[P*mec-4*::mCherry] II vs. strain ZB4902: *sem-2(n1343)* I; bzIs166[P*mec-4*::mCherry] II. Exophers were scored on Ad1 because of the damaging excessive bagging that ensues in this background. Total of three trials (50 worms per trial, each trial one dot); ****$p<0.0001$ in Cochran–Mantel–Haenszel test, as compared to the wild-type control. (**G**) Genetic interventions that induce egg retention elevate exopher levels. *lin-39* RNAi on a strain expressing P*mec-4*::mCherry from a multi-copy transgene. ALMR exopher scores Ad2. Strain ZB4757: bzIs166[P*mec-4*::mCherry] II treated with RNAi against *lin-39*. Total of six trials (50 worms per trial, each trial one dot); ****$p<0.0001$ in Cochran–Mantel–Haenszel test, as compared to the empty vector control. (**H**) Genetic interventions that induce egg retention elevate exopher levels. *lin-39* RNAi on a strain expressing P*mec-18*::mKate from a single copy transgene. ALMR exopher scores Ad2. Strain OD2984: ltSi953 [P*mec-18*::vhhGFP4::*zif-1*::operon-linker::mKate::*tbb-2* ''UTR +*Cbr-unc-119(+)*] II; *unc-119(et3)* III treated with RNAi against *lin-39*. Total of three trials (50 worms per trial, each trial one dot); ****$p<0.0001$ in Cochran–Mantel–Haenszel test, as compared to the empty vector control. (**I**) Egg-laying modulating neurotransmitters octopamine (OA) and serotonin (5-HT) influence ALMR exophergenesis levels. Data show the mean ± SEM of percentage hermaphrodite ALMR neurons exhibiting an

*Figure 3 continued on next page*

Figure 3 continued

exopher event at Ad2 after 48 hr of treatment with 20 mM octopamine (OA) or 4 mM serotonin (5-HT) on OP50 bacteria seeded NGM plates. Because 5-HT (which increases egg laying) was hypothesized to suppress exopher production, we assayed under conditions of 6 hr food limitation, which markedly raises the exopher production baseline, enabling easier quantification of suppression effects (*Cooper et al., 2021*). Total of three trials and 50 hermaphrodites per trial for each condition. ***$p<0.001$ or ****$p<0.0001$ in Cochran–Mantel–Haenszel test, as compared to the control group treated with M9 buffer (solvent for OA or 5-HT). (**J**) Mutant *goa-1(n1134),* with hyperactive egg-laying and low egg retention in the uterus (pictures on the left, representative of 20, and scale bar = 50 μm), has low exopher scores at Ad2. Strain ZB4757: bzIs166[P$_{mec-4}$::mCherry] II vs. strain ZB5352: *goa-1(n1134)* I; bzIs166[P$_{mec-4}$::mCherry] II. Boxes highlight egg zone. **$p<0.01$ in Cochran–Mantel–Haenszel test.

The online version of this article includes the following source data for figure 3:

**Source data 1.** Daily progeny count for panel A, and exopher score for panels B, D, E, F, G, H, I, J.

We found that the associated massive egg retention correlated with a dramatic elevation of exopher numbers for each genetic impediment to egg laying (*egl-9* (*Figure 3D*); *egl-3* (*Figure 3E*); *sem-2* (*Figure 3F*), *lin-39* RNAi (*Figure 3G*)). For example, under the treatment of *lin-39* RNAi, ~60% of the wild-type strain expressing mCherry in ALMR by a multi-copy transgene produced ALMR exophers on Ad1, compared to only ~7% of the same strain treated with empty vector (EV) control (*Figure 3G*). We conclude that regardless of the genetic strategy employed to trap eggs in the body, egg retention can lead to high ALMR exopher production.

In complementary studies, we examined the impact of egg retention on ALMR exophergenesis when we expressed fluorescent protein mKate from a single copy transgene (i.e. in the absence of an over-expressed reporter). We found that the empty vector control treatment without induction of egg retention is associated with no ALMR exophergenesis in the single copy mKate transgenic strain (0% in all three trials). However, when we treated with *lin-39* RNAi to induce egg retention, we measured ~60% exophergenesis (*Figure 3H*). Thus regardless of the expression levels of exogenous proteins in the ALMR neuron, the egg retention condition can induce high exopher production.

## Hyperactive egg-laying and consequent low egg retention are associated with low exopher production

Neurotransmitters octopamine (OA) and serotonin (5-HT) have been well-documented to play opposing roles in *C. elegans* egg-laying behavior (*Chase and Koelle, 2007*). Feeding *C. elegans* octopamine strongly suppresses egg-laying to induce egg retention, while supplementing with 5-HT causes hyperactive egg-laying (*Horvitz et al., 1982*). Consistent with outcomes in animals physically blocked for egg-laying, treatment with egg retention-promoting OA enhanced ALMR exophergenesis (*Figure 3I*).

To test the outcome of 5-HT-induced enhanced egg laying, we measured the impact of 5-HT consequent to 6-hr food withdrawal, a condition that we previously found markedly enhances ALMR exophergenesis (*Cooper et al., 2021*), and, therefore, is expected to increase the dynamic range for scoring exopher suppression. We find that 5-HT treatment, which lowers egg retention, strongly suppresses fasting-associated ALMR exophergenesis (*Figure 3I*). Although perturbing neuronal signaling holds complex consequences for whole-animal physiology, our findings are consistent with a model in which high egg load increases neuronal exophergenesis, and low egg retention decreases exophergenesis.

The egg-laying circuit Is controlled in part by Goα inhibition—null allele *goa-1(n1134)* removes this inhibition such that eggs are often laid at very early developmental stages (2–4 cell stage) rather than being retained in the uterus until gastrulation (~30 cell stage) (*Waggoner et al., 2000*). We introduced *goa-1(n1134)* into the mCherryAg2 background and scored ALMR exopher events at Ad2. We confirmed that *goa-1(n1134)* retains few eggs in the uterus and is associated with a significantly lower number of ALMR exopher events as compared to the age-matched wild-type control (*Figure 3J*).

In sum, manipulation of uterine egg occupancy is strongly correlated with the extent of ALMR neuronal exophergenesis—high egg retention promotes high exophergenesis and low egg retention is associated with low exophergenesis.

## ALMR neuron proximity to the egg zone correlates with exophergenesis frequency

The ALML and ALMR anterior touch neurons share developmental, morphological, and functional similarities (*Chalfie and Sulston, 1981*), yet paradoxically, ALMR consistently produces exophers at higher levels than ALML (*Melentijevic et al., 2017*; also *Figure 4G*). The ALM neurons are embedded within the *C. elegans* hypodermis, but the ALMR soma is situated in the vicinity of the gonad and its resident eggs, whereas the ALML soma, on the opposite side of the animal, is positioned closer to the intestine (*Figure 4A and B*; *Goodman, 2006*).

To ask if proximity to the gonad is correlated with exopher production, we randomly selected 128 Ad2 hermaphrodites that expressed mCherry in the touch receptor neurons, imaged in brightfield to visualize the egg zone of each animal, and imaged again in the red channel to visualize the touch neuron, recording the relative positions of ALMR and the egg zone (*Figure 4C*). We also assessed whether the ALMR neuron had produced an exopher or not.

We found that 37/128 ALMRs examined had produced an exopher (Exopher+), and that 36/37 (95%) of the Exopher+ ALMR neuronal somas that had produced exophers were positioned within the visualized egg zone (*Figure 4D*). For the Exopher- ALMRs that had not produced exophers, 63/91 (70%) had neuronal somas located in the egg zone. Thus, although ALMR soma positioning in the egg zone does not guarantee exophergenesis in the mCherryAg2 strain, the neurons that did make exophers were nearly always in the egg zone (*p*<0.01 in Chi-Square test, *Figure 4D*).

To further test for the association of egg zone proximity to ALMR and exopher production, we genetically shifted ALM position. During development, the ALM neurons migrate posteriorly to near the mid-body (*Sym et al., 1999*), and most commonly, ALM somas are situated posterior to AVM. ALM soma positions, however, can be influenced by migration and specification cues. In particular, transgenic introduction of a *mec-3* promoter fragment bearing an internal deletion (fusion of the −1 to −563 sequences to the −1898 to −2372 *mec-3* promoter fragment, plasmid pJC4) can induce anterior ALM migration during development, sometimes resulting in final ALM positions anterior to AVM (*Toms et al., 2001*; *Figure 4E*). We took advantage of the partially deleted *mec-3* promotor sequences in pJC4 to manipulate the ALM position. In these studies, we introduced pJC4 with the co-transformation marker pRF4 (*rol-6(su1006)*) that disrupts the cuticle to induce rolling of transgenic animals into the P*mec-4*::mCherry background. Rol hermaphrodites have a strikingly high baseline of ALMR exophergenesis (~40% exophers in rollers vs. ~20% in the wild-type). Strikingly, we found that when ALMs are situated anterior to AVM, ALMR exophergenesis drops to ~5% (4/73) vs 71% for the posterior position (55/78) (*Figure 4F*). Although we cannot exclude that physiological changes in differently-positioned touch neurons underlie reduced exophergenesis, data are consistent with a model in which proximity to the egg zone correlates with exophergenesis.

Another way to increase egg proximity to ALMR is to disrupt egg-laying capacity, which confers egg retention and uterine expansion. We hypothesized that in the *sem-2(rf)* mutant, which is associated with considerable internal egg accumulation, additional touch neurons should experience increased proximity to eggs in the blocked uterus. Indeed, we find that in the *sem-2(rf)* background, every touch neuron that is positioned in the general region of the expanded uterus (ALML, AVM, PVM) increases exopher production, but the posterior PLM neurons, which cannot be approached by the gonad, do not produce exophers (*Figure 4G*). Thus, touch neurons can be stimulated to produce exophers if the egg domain is artificially brought closer to them. Data are consistent with a model in which ALMR normally makes most exophers because of its closest natural positioning near the egg-filled uterus. Possibilities are that a diffusible signal may emanate from the filled uterus or that mechanical pressure associated with a filled uterus might signal enhanced exophergenesis.

## Uterine expansion associated with high egg load correlates with high exophergenesis

How might the presence of eggs signal to the maternal neurons to induce exophers? We consider two main possibilities: (1) eggs filling the uterus might exert physical pressure that activates essential stretch-signaling for young adult neuronal exopher release. This mechanical stretch signal might act directly (for example introducing chronic and/or dynamic pressure on the touch neurons), or indirectly (possibly inducing the stretched uterus/somatic gonad to release chemical signals that promote neuronal exopher formation); (2) early fertilized eggs might release a short-range chemical signal that

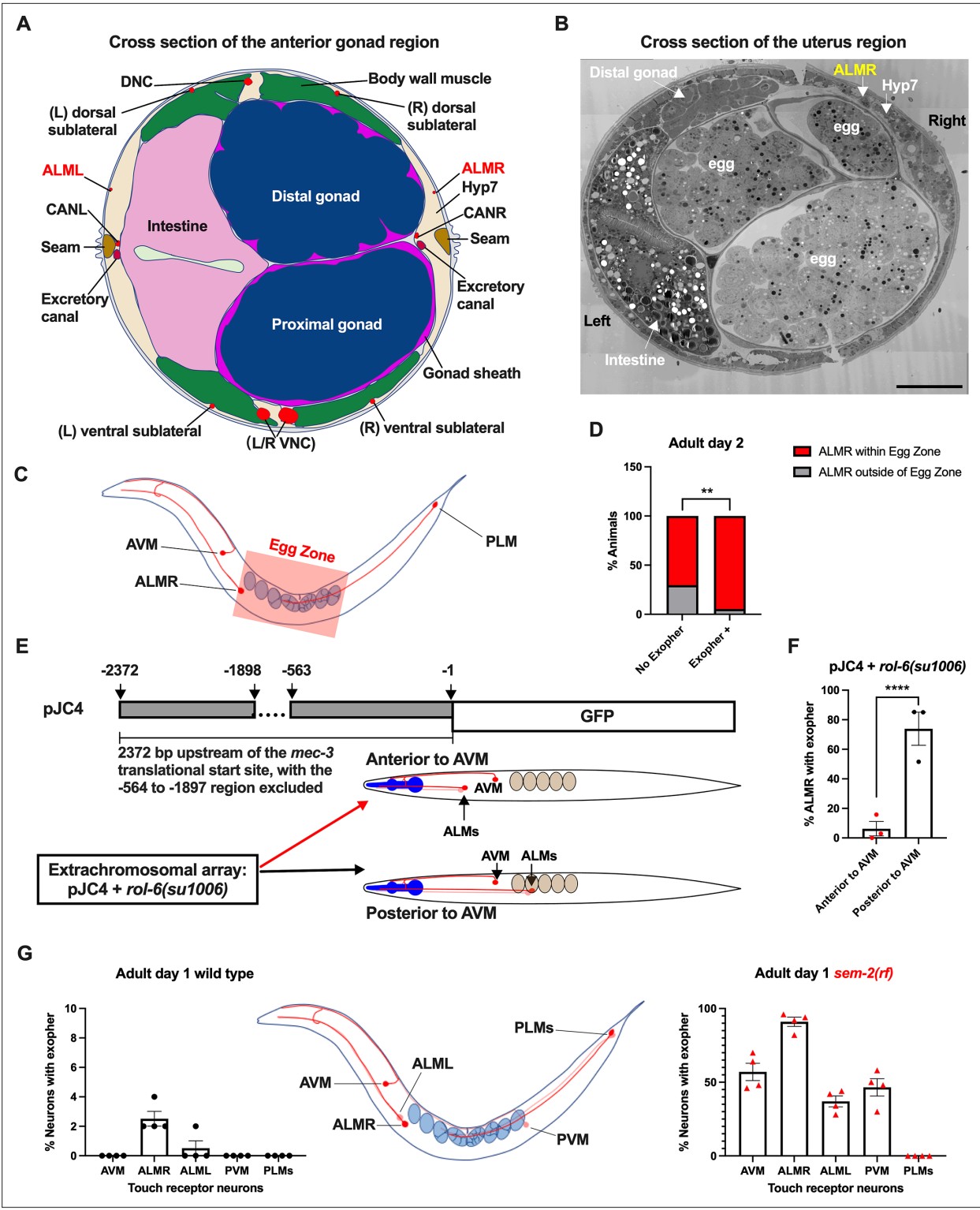

**Figure 4.** Anterior Lateral Microtubule cell (ALMR) exophergenesis levels correlate with proximity to the egg zone. (**A**) ALMR is positioned close to the uterus and Anterior Lateral Microtubule cell (ALML) is situated on the opposite side, close to the intestine. A diagram of ALMR and ALML positions relative to major body organs in cross-section that is anterior to the 'egg zone.' Image drawing based on the EM pictures of adult hermaphrodite slice #273 of WormAtlas. DNC: dorsal nerve cord; VNC: ventral nerve cord; Hyp7: hypodermal cell 7; CAN(L/R): canal-associated neurons (left/right); neurons in red. (**B**) Electron microscopy cross-section image of the uterus region indicating ALMR soma and eggs within the adult uterus. Note ALMR is close to the egg-filled uterus, ALML is on the opposite side closer to the intestine. The ALML soma is not evident in this cross section. Scale bar = 10 μm. (**C**) Illustration of the egg zone definition, distance between outermost eggs. The measure of this distance corresponds to uterine length. (**D**) ALMRs

*Figure 4 continued on next page*

*Figure 4 continued*

positioned close to the filled gonad produce exophers more frequently than ALMRs that are positioned a bit more distally. We selected Ad2 mCherry animals at random, then identified whether or not ALMR had produced an exopher, and subsequently determined whether ALMR was positioned within the egg zone or outside the egg zone as indicated (neuronal soma positioning differences are a consequence of developmental variation). Neurons with somas positioned further from the egg area produced fewer exophers than neurons within the egg zone indicated; total of 91 worms for the 'No Exopher' group and 37 for the 'Exopher +' group; **$p<0.01$ in Chi-Square test. (**E**) The Aamodt group (**Toms et al., 2001**) previously reported that high copy numbers of plasmid pJC4 containing the *mec-3* promoter region (−1 to −563, and −1898 to −2372 of the *mec-3* translational start) exhibited increased abnormal positioning of ALM neurons anterior to AVM. We introduced plasmid pJC4 along with transformation reporter pRF4 *rol-6(su1006)* in the background of mCherryAg2 (note this revealed that *rol-6(su1006)* is a strong exopher enhancer) and identified neurons that were positioned posterior to AVM (normal, close to the uterus) and those that were positioned anterior to AVM (further away from the uterus). (**F**) ALMR neurons genetically induced to adopt positions further away from the uterus generate fewer exophers then those close to the uterus. We counted numbers of exophers produced in Ad2 Rol hermaphrodites for each position type. Strain ZB5046: Ex [(pJC4) P$_{mec-3}$::GFP +pRF4]; bzIs166[P$_{mec-4}$::mCherry] II. Total of three trials (61(34a + 27p); 39(19a + 20p); 51(20a + 31p) animals per trial); ****$p<0.0001$ in Cochran–Mantel–Haenszel test. (**G**) When eggs cannot be laid in the *sem-2(rf)* mutant, the eggs that accumulate in the body are brought in closer proximity to ALML, AVM, and PVM touch neurons with a resulting increase in their exopher production. Strain ZB4757: bzIs166[P$_{mec-4}$::mCherry] II vs. strain ZB4902: *sem-2(n1343)* I; bzIs166[P$_{mec-4}$::mCherry] II. Left, Exopher scoring (Mean ± SEM) of all six touch receptor neurons in Ad1 wild-type hermaphrodite; Right, Exopher scoring (Mean ± SEM) of all six touch receptor neurons in Ad1 *sem-2(rf)* hermaphrodite. Total of 4 trials (50 worms per trial) for each. Wild-type, egg laying proficieint animals on Ad1 exhibit low exopher levels, but when eggs accumulate early in the *sem-2* mutant, exophers markedly increase in ALMR and other touch neurons that are in the vicinity of an expanded uterus. PLM neurons are situated posterior to the anus and are not subject to uterine squeezing effects.

The online version of this article includes the following source data for figure 4:

**Source data 1.** Contingency data for panel D, and exopher score for panels F, G.

contributes to young adult proteostasis reorganization (**Labbadia and Morimoto, 2014**; **Labbadia and Morimoto, 2015**) to promote exophergenesis.

To begin to dissect the role of egg pressure in promoting exophergenesis, we analyzed the physical relationships of neurons, eggs, and uterine shape. Eggs can readily be observed to distort tissue structure in young adult *C. elegans*. For example, in a strain that expresses GFP to label the hypodermis and expresses mCherry to label the touch neurons, the distortion of the hypodermis by eggs can be easily visualized as dark non-fluorescent eggs project through the observation plane of the hypodermis (**Figure 5A**). Thus, the uterus can approach and pressure surrounding tissue, including touch neurons.

We quantitated the absolute uterine length as an indicator for stretch in relation to exopher production levels under representative conditions of high and low exophergenesis. We found that egg retention mutants that exhibit high exophergenesis, *egl-3(Δ)* (**Figure 5B**), *egl-9(Δ)* (**Figure 5C**), and *sem-2(rf)* (**Figure 5D**), had significantly longer egg zones (i.e. uterus length) as compared to wild-type. In contrast, *cbd-1* RNAi (**Figure 5E**), sperm-less induction with SPE-44 AID (**Figure 5F**), and hyperactive egg-laying mutant *goa-1(Δ)* (**Figure 5G**), which we find to be strong exophergenesis suppressors, are all associated with short uterine egg zones. Knocking down either the 4 cell stage gene *mex-3* or the gastrulation gene *gad-1*, which are associated with neither egg retention nor exopher elevation, does not have an extended egg zone/uterine length (**Figure 5H**), so not all developmental compromises are associated with uterine extension.

## Even when eggshell production and early embryonic divisions are disrupted, forced uterine expansion can elevate exopher levels

Our initial studies suggest extended uterine length is correlated with high exopher levels, but high egg retention is also a feature of an extended uterus. To begin testing a requirement for fertilized eggs per se in the exopher influence, we asked whether egg viability is essential for promoting early adult exophergenesis. We manipulated egg integrity/uterine contents in egg retention mutants by egg/embryo perturbation, testing for impact on exophergenesis.

Under conditions of *cbd-1* RNAi, eggshell development and embryonic development are blocked; eggs that form are fragile and can be malformed consequent to passing through the spermatheca into the uterus; embryonic development does not proceed and egg remnants tend to be sticky (**Johnston et al., 2010**). As shown in **Figure 2B** (and again in **Figure 6A**), treating WT reproductive animals (that have functional egg-laying capacity) with *cbd-1* RNAi to kill embryos exerts a potent block on ALMR exophergenesis. We proceeded to test the consequence of *cbd-1* RNAi in mutants that cannot

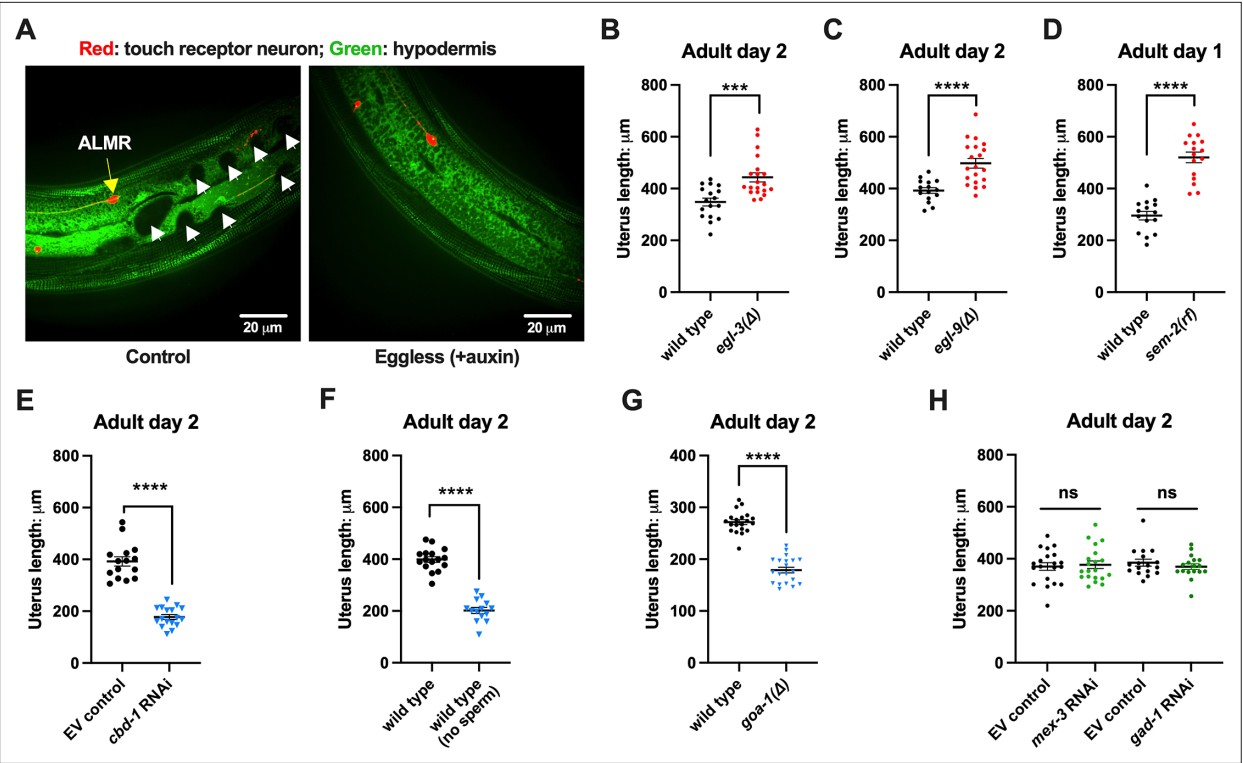

**Figure 5.** Uterine length measures for egg-laying defective and gastrulation defective mutants support the correlation of high exopher production and uterine expansion. (**A**) Eggs can distort tissues in their vicinity. Shown (representative of 10, scale bar = 20 μm) is strain ZB4942: fxIs1[P*pie-1*::TIR1::mRuby] I; bzIs166[P*mec-4*::mCherry] II; *spe-44(fx110[spe-44*::degron]) IV; pwSi93[Phyp7::oxGFP::*lgg-1*], with touch neurons expressing mCherry (red); and the hypodermis expressing GFP. Dark round areas (white arrows) are eggs that press into the hypodermis when viewed in this focal plane. On the right, slit-like dark regions correspond to hypodermal seam cells. (**B**) Uterus length of WT vs. *egl-3(Δ)*; strain ZB4757: bzIs166[P*mec-4*::mCherry] II vs. ZB4904: bzIs166[P*mec-4*::mCherry] II; *egl-3(gk238)* V. n = ~20 hermaphrodites from one trial. ***p<0.001 in two-tailed *t*-test. Note that we did not normalize uterine length to body length in B-E. (**C**) Uterus length of wild-type (WT) vs. *egl-9(Δ)* ZB4757: bzIs166[P*mec-4*::mCherry] II vs. ZB4772: bzIs166[P*mec-4*::mCherry] II; *egl-9(sa307)* V. n = ~20 hermaphrodites from one trial. ****p<0.0001 in two-tailed *t*-test. (**D**) Uterus length of WT vs. *sem-2(rf)*; strain ZB4757: bzIs166[P*mec-4*::mCherry] II vs. ZB4902: *sem-2(n1343)* I; bzIs166[P*mec-4*::mCherry] II. n = ~20 hermaphrodites from one trial. ****p<0.0001 in two-tailed *t*-test. (**E**) Uterine length is short under *cbd-1* RNAi compared to WT + empty vector RNAi. Uterus length of strain ZB4757: bzIs166[P*mec-4*::mCherry] treated with RNAi against *cbd-1* or control empty vector feeding RNAi. n = ~20 hermaphrodites from one trial. ****p<0.0001 in two-tailed *t*-test. (**F**) When sperm maturation is blocked in egg-laying competent animals, leaving oocytes to occupy reproductive structures, the uterus length is short. Uterus length of strain ZB4749: fxIs1[P*pie-1*::TIR1::mRuby] zdIs5[P*mec-4*::GFP] I; bzIs166[P*mec-4*::mCherry] II; *spe-44(fx110[spe-44*::degron]) IV. 1 mM auxin treatment induces the no sperm status. n = ~20 hermaphrodites from one trial. ****p<0.0001 in two-tailed *t*-test. (**G**) Uterine length is short in the hyperactive egg-laying mutant which has low occupancy of eggs in the uterus. Uterus length of strain N2: wild-type vs. strain MT2426: *goa-1(n1134)* I. n=20 hermaphrodites from one trial. ****p<0.0001 in two-tailed *t*-test. (**H**) Knocking down the 4 cell stage gene *mex-3* or the gastrulation gene *gad-1* has normal uterine length. Uterus length of strain ZB4757: bzIs166[P*mec-4*::mCherry] treated with RNAi against the *mex-3* or *gad-1* gene. *mex-3* RNAi disrupts embryonic deveopment at the 4 cell stage, while *gad-1* RNAi disrupts gastrulation at the stage at which eggs are normally laid and perturbs later development but not egg shell formation and egg laying. n = ~20 hermaphrodites from one trial. Not significant (ns) in two-tailed *t*-test as compared to the empty vector control.

The online version of this article includes the following source data for figure 5:

**Source data 1.** Uterus length data for panels B, C, D, E, F, G, H.

extrude eggs or their remnants, and, therefore, would retain defective eggshell/dead embryos in the uterus.

We subjected the *sem-2(rf)* egg-laying defective mutant to *cbd-1* RNAi so that the uterus would fill with defective eggshell/dead embryo remains. Strikingly, we found that *cbd-1* RNAi/dead embryo retention in the *sem-2(rf)* background is still associated with significantly elevated levels of ALMR exophergenesis (~45%), while in egg-laying proficient wild-type, barely any ALMR exophergenesis is observed under *cbd-1* RNAi conditions (<1%) (69/150 for *sem-2* egg-laying blocked *cbd-1* RNAi vs. 1/150 for egg-laying proficient *cbd-1* RNAi, *Figure 6A*).

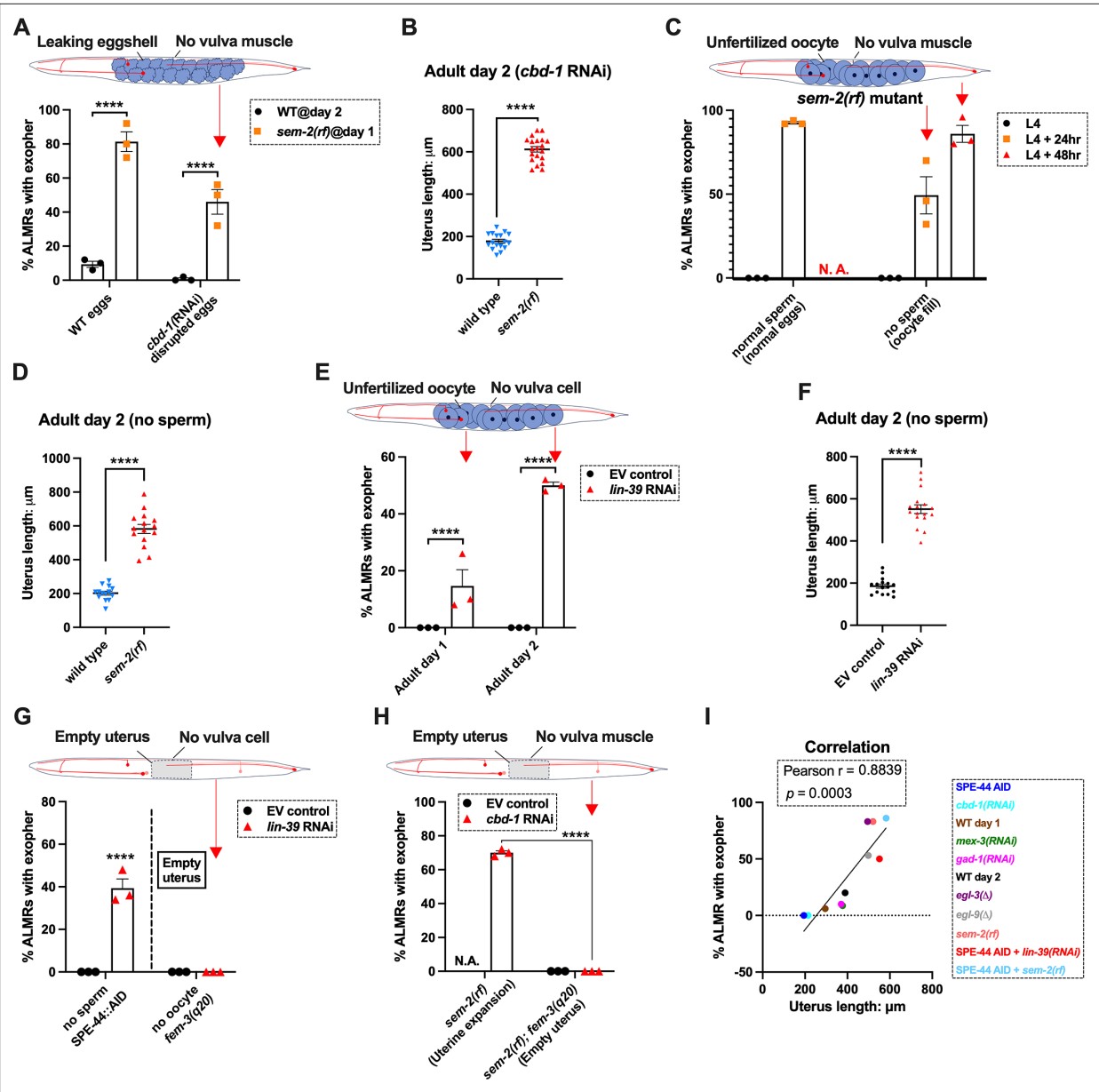

**Figure 6.** Uterine expansion correleates strongly with Anterior Lateral Microtubule cell (ALMR) exophergenesis regardless of whether eggs, oocytes, or debris are retained. (**A**) Despite *cbd-1* RNAi mediated disruption of eggshell formation and earliest embryonic cell divisions, exopher levels are high in the *sem-2(rf)* egg retention background. The percentage of ALMR exopher events among 50 Ad2 wild-type (left) or Ad1 *sem-2(rf)* hermaphrodite *C. elegans* that are treated with either empty vector control RNAi or RNAi targeting *cbd-1* in each trial (total of 3 independent trials). *sem-2* mutants bag extensively at Ad2 and cannot be tested. Diagram indicates uterine filling status of test *sem-2(rf)* + *cbd-1* RNAi. Strain ZB4757: bzIs166[P*mec-4*::mCherry] II vs. strain ZB4902: *sem-2(n1343)* I; bzIs166[P*mec-4*::mCherry] II. ****p<0.0001 in Cochran–Mantel–Haenszel test. (**B**) Uterine length remains long in the egg-laying defective *sem-2(rf)* + *cbd-1* RNAi. Uterus length of strain ZB4757: bzIs166[P*mec-4*::mCherry] II vs. ZB4902: *sem-2(n1343)* I; bzIs166[P*mec-4*::mCherry] II treated with RNAi against *cbd-1*. n = ~20 hermaphrodites from one trial, Ad2, ****p<0.0001 in two-tail unpaired *t*-test. (**C**) Blocking sperm maturation in a *sem-2(rf)* mutant, which fills the uterine space with oocytes, induces exophers in the absence of eggs. The percentage of ALMR exopher events among 50 *sem-2(rf)* hermaphrodite *C. elegans* that express the SPE-44 AID system ('control' is treated with 0.25% ethanol vehicle and 'no sperm' is treated with 1 mM auxin in 0.25% ethanol from egg to adult day 2) in each trial (total of 3 independent trials). L4 stage is the last larval stage before adult. Diagram indicates uterine oocyte filling status of test *sem-2(rf)* + SPE-44 AID. Strain ZB4953: *sem-2(n1343)* fxIs1[P*pie-1*::TIR1::mRuby] I; bzIs166[P*mec-4*::mCherry] II; *spe-44(fx110[spe-44::degron])* IV. (**D**) When sperm maturation is disrupted in mutants blocked for egg laying, leaving oocytes to occupy reproductive structures, the uterus expands as oocytes accumulate. Uterus length of strain ZB4749: fxIs1[P*pie-1*::TIR1::mRuby] zdIs5[P*mec-4*::GFP] I; bzIs166[P*mec-4*::mCherry] II; *spe-44(fx110[spe-44::degron])* IV vs. ZB4953: *sem-2(n1343)* fxIs1[P*pie-1*::TIR1::mRuby] I; bzIs166[P*mec-4*::mCherry] II; *spe-44(fx110[spe-44::degron])* IV. 1 mM auxin treatment induces the no sperm status to both strains. n = ~20 hermaphrodites from one trial, Ad2. ****p<0.0001 in two-tail unpaired *t*-test. (**E**) Disrupting sperm maturation in *lin-39* RNAi animals blocked for egg-laying fills the uterine space with oocytes, and induces

*Figure 6 continued on next page*

*Figure 6 continued*

exophers in the absence of eggs. The percentage of ALMR exopher events among 50 adult day 1 or 2 SPE-44 AID no sperm hermaphrodite *C. elegans* treated with either control RNAi or *lin-39* RNAi in each trial (total of 3 independent trials). Diagram indicates uterine oocyte filling status of test *lin-39* RNAi + SPE-44 AID. ZB4749: fxIs1[P$_{pie-1}$::TIR1::mRuby] zdIs5[P$_{mec-4}$::GFP] I; bzIs166[P$_{mec-4}$::mCherry] II; spe-44(fx110[spe-44::degron]) IV. ****$p$<0.0001 in Cochran–Mantel–Haenszel test. (**F**) When sperm maturation is blocked, leaving oocytes to occupy reproductive structures, the uterus length is short; but if oocytes cannot be laid in the *lin-39* RNAi background, the uterus expands as oocytes accumulate. Uterus length of strain ZB4749: fxIs1[P$_{pie-1}$::TIR1::mRuby] zdIs5[P$_{mec-4}$::GFP] I; bzIs166[P$_{mec-4}$::mCherry] II; spe-44(fx110[spe-44::degron]) IV +1 mM auxin treatment to eliminate sperm maturation. n = ~20 hermaphrodites. ****$p$<0.0001 in two-tail unpaired *t*-test. (**G**) Blocking oocyte production in the background of *lin-39* RNAi-mediated disruption of the egg-laying apparatus eliminates early adult exophergenesis. We used SPE-44 AID to block sperm maturation and *fem-3(q20)* to prevent oocyte production; *lin-39* RNAi to disrupt egg-laying capacity. Diagram indicates empty uterus status of test *fem-3(q20); spe-44 AID; lin-39(RNAi)* strain. Exopher scoring of Ad2 ZB4749: fxIs1[P$_{pie-1}$::TIR1::mRuby] zdIs5[P$_{mec-4}$::GFP] I; bzIs166[P$_{mec-4}$::mCherry] II; spe-44(fx110[spe-44::degron]) IV +1 mM auxin or ZB5042: bzIs166[P$_{mec-4}$::mCherry] II; fem-3(q20)ts IV, treated with either control empty vector (EV) RNAi or *lin-39* RNAi at 25 °C. Total of three trials (50 worms per trial). ****$p$<0.0001 in Cochran–Mantel–Haenszel test. (**H**) Blocking oocyte production in the background of *sem-2(rf)*-mediated disruption of the egg-laying apparatus eliminates early adult exophergenesis. We used *cbd-1* RNAi to disrupt eggshell and *fem-3(q20)* to prevent oocyte production; *sem-2(n1343)* to disrupt egg-laying capacity. Exopher scoring of adult day 2 hermaphrodites, treated with either control empty vector (EV) RNAi or *cbd-1* RNAi at 25 °C. Total of three trials (50 worms per trial). Diagram indicates empty uterus status of test *fem-3(q20); cbd-1 RNAi; sem-2(rf)* strain. (**I**) The uterus length is correlated with ALMR exophergenesis. Data shown are the mean of uterus length (X-axis) and percentage ALMRs with exopher (Y-axis) for different genotypes/treatments measured in this study. The correlation line is based on a linear fit model and the Pearson r and *p* value is based on the correlation assay. Uterus length from short to long: SPE-44 AID; *cbd-1* RNAi; wild-type (adult day 1); *mex-3* RNAi; *gad-1* RNAi; wild-type (adult day 2); *egl-3(Δ); egl-9(Δ); sem-2(rf)*; SPE-44 AID + *lin-39* RNAi; SPE-44 AID + *sem-2(rf)*.

The online version of this article includes the following source data and figure supplement(s) for figure 6:

**Source data 1.** Exopher score for panels A, C, E, G, H, I, and uterus length data for panels B, D, F, I.

**Figure supplement 1.** Representative pictures of oocytes retention (red rectangle) in the uterus of Adult day 2 hermaphrodite.

Importantly, under conditions of defective eggshell/dead embryo retention associated with *cbd-1* RNAi; *sem-2(rf)*, the uterine egg zone is expanded (**Figure 6B**), extending the correlation of exopher production with uterine length. We conclude that intact eggshells and earliest embryonic divisions are not required for the boost in exopher production observed when uterine contents are forced to accumulate—uterine retention of dead eggs and egg remnants is sufficient for exopher elevation if the egg laying apparatus is defective. Expansion of the uterine compartment, rather than eggshell/embryo integrity, tracks with exopher elevation.

## Forced uterine expansion via oocyte accumulation can elevate exopher levels

Although uterine retention of malformed inviable embryos is sufficient to elevate neuronal exophers when egg laying is blocked, the defective *cbd-1* embryos or debris might still release egg-associated chemical signals. To test for a requirement of any fertilization-dependent egg signals in the egg-laying compromised mutants, we asked whether uterine filling with only oocytes can suffice to promote neuronal exopher elevation. Unfertilized oocytes cannot initiate embryonic development or egg-shell biosynthesis; nor can oocytes elevate ALMR exophergenesis in hermaphrodites that are proficient at egg-laying (**Figure 1G&H**).

We tested two distinct uterine retention conditions—*sem-2(rf)* and *lin-39* RNAi—in which we used the auxin-inducible degron system to disrupt sperm maturation such that only oocytes filled the gonad. Note that in the absence of sperm, oocytes do not mature but are 'ovulated' at a reduced rate compared to in the presence of sperm; oogenesis continues such that ~25 oocytes are typically found stacked in the gonad (**Figure 6—figure supplement 1**; **McGovern et al., 2007**). For both genetic retention strategies, we found that build-up of retained oocytes in egg-laying blocked animals was both sufficient to elevate exophers (**Figure 6C&E**) and expand the uterus (**Figure 6D&F**; **Figure 6—figure supplement 1**). Moreover, the oocyte retention was similarly efficacious in elevating exopher production to egg retention, increasing ALMR exophergenesis to approximately 80% in the *sem-2(rf)* mutant (**Figure 6C**). We conclude that fertilization, egg shells, and egg remnants are not essential for the early adult exopher peak. Expansion of the uterus with unfertilized oocytes can suffice to elevate neuronal exopher formation.

## Lack of a functional egg-laying apparatus does not induce exopher elevation when the uterus is not filled

The above-described experiments left open the possibility that the lack of a functional egg-laying apparatus itself might be causative in the elevation of exopher production. To address this possibility, we compared disruption of sperm (permissive for oocyte accumulation) to disruption of oogenesis (effectively empties the uterus) when egg-laying capacity was compromised by *lin-39* RNAi (*Figure 6G*). *lin-39* RNAi + oocyte retention promotes exopher formation, but eliminating oocytes (*fem-3(gf)*) eliminates exopher elevation even when egg-laying is blocked by *lin-39* RNAi. That is to say, although oocyte accumulation with uterine expansion suffices to elevate exophers, removing the oocytes and uterus occupancy eliminates the exopher boost. We observe the same outcome of suppressed exopher formation when *cbd-1* RNAi-induced dead embryo retention in the *sem-2(rf)* egg-laying defective mutant (which is exopher-inducing) is prevented from oocyte production by *fem-3(gf)* (*Figure 6H*). Thus, disruption of egg laying on its own is not the driving factor in high exophergenesis; rather, uterine filling is required.

We revisited the relationship of uterine length and exopher level by adding data from the studies with oocyte retention to reinforce the conclusion that ALMR exophergenesis is strongly correlated with the level of uterus stretching caused by the accumulation of uterine contents (*Figure 6I*).

## Sustained physical distortion of the gonad by fluid injection can rapidly elevate exopher production

To independently test for a role in physical stretch/filling of the uterus in exopher induction, we distorted the gonad compartment by injecting dye-containing M9 buffer into a very young adult. The animal subjects we tested were vulva-less (*lin-39* RNAi) and also subjected to SPE-44 AID to block sperm production. These vulva-less + sperm less hermaphrodites normally exhibit high ALMR exophergenesis at late Ad1 and Ad2 (*Figure 6C&D*) due to oocyte accumulation. To avoid oocyte influence, we conducted our physical expansion studies just as animals reached Ad1, a time when ALMR exophergenesis is typically not observed (*Figure 7A*).

We picked the L4 stage vulva-less (*lin-39* RNAi)+ sperm less (SPE-44 AID) hermaphrodites for age synchronization (20 °C, grown for 12 hr after L4 selection) and then injected dye containing M9 buffer into the uteri of these very young adult days 1 hermaphrodites, scoring for exopher production ~10 min after the injection (*Figure 7B*).

Control animals (mock-injected animals that were stabbed without fluid delivery) exhibited no ALMR exophers. By contrast, we found that when we held injection pressure continuously for 2 or more minutes, ~20% of the ALMs scored exhibited an exopher event shortly after injection (*Figure 7C&D*). Injection experiments using animals with functional vulvae (in which injected material is rapidly extruded through the vulval opening) failed to induce ALMR exophers (*Figure 7E*), supporting that ALMR exophergenesis caused by 2 min injection of dye-containing M9 is due to the physical distortion of gonad rather than the chemical impact of the dye-containing M9.

In sum, exopher induction by uterine accumulation of eggs, malformed eggs, dead embryos, oocytes, or fluid-induced expansion supports a model in which early adult ALMR exophergenesis is elevated by physical distortion of the uterus that occurs with reproduction (*Figure 7F*).

## Discussion

Exopher production by proteo-stressed *C. elegans* touch neurons occurs with a striking temporal pattern that features an early adult peak of exophergenesis coincident with the period of maximal egg production. Here, we report that the early exophergenesis peak is dependent on uterine occupation, which normally is conferred by fertilized egg accumulation prior to the deposition of eggs. Uterine expansion that is associated with the filling of a blocked uterus with unfertilized oocytes or by fluid alone can also induce high neuronal exopher production, supporting a model in which the physical distention associated with uterine occupancy, rather than chemical signals derived from fertilized eggs per se, is a required component of the signaling relay between reproduction and young adult neuronal debris elimination. Trans-tissue cross-talk to the maternal nervous system thus appears accomplished via mechanical force transduction. Our findings hold implications for mechano-biology in neuronal proteostasis management.

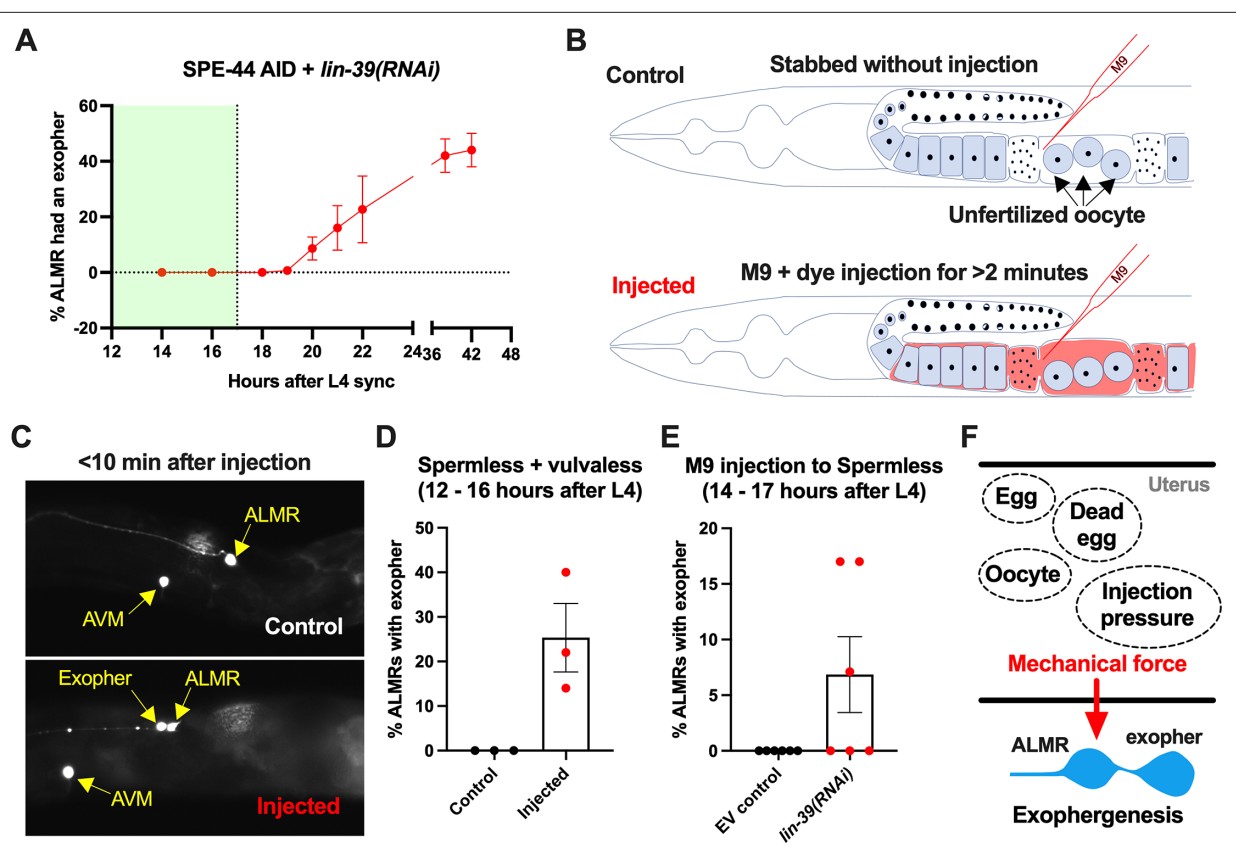

**Figure 7.** Anterior lateral microtubule cell (ALMR) exophergenesis can be induced by uterine compartment distortion that accompanies fluid injection. (**A**) Summary of timing of ALMR exophergenesis from age synchronized spermless + vulvaless hermaphrodite. Total of three trials and 50 hermaphrodites in each trial. Strain ZB4749: fxIs1[P$_{pie-1}$::TIR1::mRuby] zdIs5[P$_{mec-4}$::GFP] I; bzIs166[P$_{mec-4}$::mCherry] II; *spe-44(fx110[spe-44::degron])* IV, cultured on the 1 mM auxin treated nematode growth media (NGM) agar plates seeded with HT115 *E. coli* expressing dsRNA against the *C. elegans lin-39* gene. Eventually, oocytes accumulate in this strain, but at worst very few are evident in the timeframe in which we performed injections. Data shown are mean ± SEM at each time point. The data demonstrate that there is no ALMR exophergenesis in this background before the 18th hr after L4 sync. Testing the impact of injection on ALMR exophergenesis before the 17th hr after L4 thus monitors injection consequences during a timeframe in which no exophers are normally produced. (**B**) Illustrated experimental design for testing the ALMR exophergenesis response to physically expanding the gonad via 2 min continuous fluid injection. We performed 2+ min duration injections of M9 buffer mixed with food color dye 1:10 or 1.5:10 dye/M9 ratio (to verify successful injection; dye contains water, propylene glycol, FD&C reds 40 and 3, propylparaben) into the uteri of sperm-less only (EV control) or sperm-less + vulvaless (treated with *lin-39* RNAi) animals. Strain: ZB4749 as in panel A, treated with either empty vector (EV) or RNAi against *lin-39* in the presence of 1 mM auxin. *lin-39* RNAi disrupts vulval development such that injected fluids are retained to expand the uterus. In injections with animals that have functioning egg-laying apparatus, fluids can exit the animal and do not expand the uterus. (**C**) 2 min sustained injection into egg-laying blocked, reproduction blocked animals can induce rapid exophergenesis: representative picture of an ALMR exophergenesis event consequent to injection. (**D**) 2 min sustained injection into egg-laying blocked, reproduction blocked animals can induce rapid exophergenesis: exopher scoring of the control mock injected and 2 min injected animals. Strain ZB4749: fxIs1[P$_{pie-1}$::TIR1::mRuby] zdIs5[P$_{mec-4}$::GFP] I; bzIs166[P$_{mec-4}$::mCherry] II; *spe-44(fx110[spe-44::degron])* IV treated with auxin and *lin-39* RNAi to induce the sperm-less and vulva-less status, respectively. Data represent a total of three trials (6–10 worms in each trial). *p<0.05* in Cochran–Mantel–Haenszel test, as compared to the no-injection fluid control. (**E**) 2 min sustained injection induces rapid ALMR exophergenesis only from vulvaless animals. Data showing the exopher scoring of the sperm-less EV control (with functional vulva) or sperm-less +vulvaless (treated with *lin-39* RNAi) worms. Strain: ZB4749 (genotype in panel A legend) treated with either EV or RNAi against *lin-39* in the presence of 1 mM auxin to disrupt sperm maturation. Data represent a total of 6 trials (6–10 worms in each trial). 3 out of 6 trials showed ALMR exopher induction by M9 injection to the vulvaless worms; while not a single trial produced ALMR exopher induction by M9 injection to animals with WT vulvae. If the vulva is intact, injected fluids are observed to leak out, consistent with the assumption that the gonads of egg-laying proficient animals will not sustain required expansion. (**F**) Summary. Eggs, dead egg accumulation, oocyte accumulation, or injection pressure all lead to ALMR exophergenesis. These varied interventions have a similar impact on the uterus, which is the uterine distortion by mechanical forces. We propose that early adult ALMR exophergenesis requires mechanical or stretch-associated force generated by the uterine cargos.

The online version of this article includes the following source data for figure 7:

**Source data 1.** Exopher score for panels A, D, E.

## The mechanical landscape of reproduction that influences neuronal exophergenesis

*C. elegans* reproduction features physical expansion and contraction of multiple tissue/cell types--the gonad houses the expanding germline, and the spermatheca expands and retracts vigorously as each mature oocyte enters and exits. The uterus also stretches to house eggs and can contract locally as eggs transit or are expelled via the action of the vulval and uterine muscles. The filled reproductive apparatus can thus clearly exert both constitutive and sporadic pressure on surrounding tissues as it enacts its essential functions.

What is the source of the force that promotes exopher production? Elegant work on the egg-laying circuit (comprising the somatic gonad, the HSN and VC neurons, and the vulval, and uterine muscle) has provided evidence for mechanical signaling within the egg-laying circuit that regulates initiation, promotion, and termination of egg-laying (*Medrano and Collins, 2023*; *Ravi et al., 2018*; *Kopchock et al., 2021*; *Ravi et al., 2021*). Working backward, exopher-promoting force seems unlikely to derive from the vulva or vulval muscle contractions, since when these cells are genetically disrupted in *lin-39* RNAi or in *sem-2* mutants, high levels of exophers still are generated. Changes in spermatheca volume, which expands and contracts dramatically as mature oocytes enter via valve opening/closure (*Tan and Zaidel-Bar, 2015*), might be sensed, but spermatheca contractions are reported to be normal in hyperactive egg-laying *goa-1* (*Govindan et al., 2006*), which is an exopher-suppressing background. Under no-sperm conditions, oocyte transit rates are lower than for fertilized eggs, and sperm-derived signals influence spermathecal valve opening (*McGovern et al., 2007*; *McCarter et al., 1999*; *Miller et al., 2001*), but if the egg-laying apparatus is genetically compromised and oocytes accumulate in the absence of sperm, exopher levels are high, suggesting deficits in spermatheca operations or sperm signals per se do not drive exophergenesis. Given that under normal reproductive conditions of egg-laying proficiency, correctly shelled eggs are required for the early peak in exopher production, a plausible hypothesis was that fertilized eggs might produce an essential diffusible factor that stimulates neuronal exopher-genesis. However, exophers can be produced abundantly in the absence of fertilized eggs when the vulva is unable to open and release uterine contents, resulting in uterine distention due to debris filling. Thus, the simplest model we envision for the reproductive cues that influence maternal neuronal exophergenesis is that a filled uterus (under normal conditions the consequence of hard-shelled eggs that occupy it) is sensed and required for the early adult peak in exopher production.

## How might force be sensed and transduced?

Mechanotransduction is the sensing of a mechanical signal, such as pressure or stretch, and conversion into a cellular response. Members of several ion channel families have been implicated in sensing of touch, hearing, shear stress, and pressure, including Piezo, TRP, and DEG/ENaC families (*Delmas and Coste, 2013*). These are rational candidates for mechanosensors in neurons, the uterus, or other cells that might act in a relay between the uterus and neuron.

At the same time, classic mechanosensory channels have extremely rapid gating and might not be the best-suited candidates for acting in the sustained and locally dynamic forces anticipated for the reproductive uterine environment. Adhesion G-Protein Coupled Receptors, which have extracellular adhesion motifs and seven transmembrane domains characteristic of the GPCR class (*lat-1*, *lat-2*, *cdh-6* in *C. elegans* [*Mee et al., 2004*; *Willson et al., 2004*; *Hutter et al., 2000*]), or components of the YAP/TAZ transcriptional program (*Panciera et al., 2017*) may integrate responses to forces transmitted via the cytoskeleton and could be considered as potential players in the required signaling.

Determination of the identity of mechanotransducers and assignment of the site of action to the neuron, the uterus, or an intermediate relay cell type remains for future studies. Modeling will also need to incorporate the fact that fluid injections, which require 2 min long sustained application of the filling stimulus to induce exophers, could provoke exopher production on a rapid timeframe, typically recorded only 10 min after the injection period. Thus the proteostressed touch neurons appear poised to eliminate contents upon mechanical stimulation.

## Mechanical signaling in reproduction across species

Uterine stretch may be a more prevalent mechanism for inducing maternal nervous system response than currently appreciated. Distention of the female fly reproductive tract by egg passage through the

tract (normal biology) or by artificial means (experimental fluid injection) can induce behavioral attraction of the mother to acetic acid, thought to signal a favorable food environment for offspring (*Gou et al., 2014*). In this case, DEG/ENaC channel family member PPK1 expressed in a subset of mechanosensitve neurons that tile the reproductive tract and respond to its contraction/distention is required. The pathway to the behavioral change remains to be determined. Uterine stretch in mammals has also been to reported influence maternal behavior (*Kristal, 2009*).

## Why link exophers to reproduction?

Turek et al. report that exophers produced by *C. elegans* muscle cells follow a similar time course of highest production at adult day 2, and demonstrated a dependence of the temporal muscle exopher-genesis pattern on eggs, and commonly with highest exopher production in muscles in the vicinity of the uterus (*Turek et al., 2021*) (muscle exophers may be released to supply nutrients to developing progeny). Together with our observations, data raise the possibility that the onset of reproduction and the initial filling of the uterus triggers, or generates a 'license' for EV/exopher production across tissues. In the case of stressed touch receptor neurons that have been our focus, evidence suggests that deleterious protein aggregates and/or excess proteins and organelles are handed off to neighboring glial-like hypodermal cells for degradation (*Wang et al., 2023*). Clearing the nervous system (and other organs) for optimal function might confer a selective advantage for successful maternal reproduction.

In this regard, it is fascinating that the peak exopher production period is coincident with a proteostasis reconfiguration that has been well documented to accompany reproduction onset in young adult *C. elegans*. In brief, during larval development, *C. elegans* exhibits high activity of HSF-1 and consequently, HSF-1-dependent chaperone expression, but HSF-1 activity is turned down in adult life (*Labbadia and Morimoto, 2014*; *Labbadia and Morimoto, 2015*; *Sala and Morimoto, 2022*; *Sala et al., 2020*). At the same time in early adult life, proteasome activity is relatively enhanced (at least as measured in the hypodermis) (*Liu et al., 2011*). These measures may reflect a general proteostasis reorganization (chaperone activity, proteosome activity) that occurs in early adult life in response to reproduction (*Labbadia and Morimoto, 2014*; *Sala and Morimoto, 2022*). Our observations on neuronal exophers suggest that exopher-mediated content elimination may constitute another co-regulated branch of this proteostasis reconfiguration. Importantly, the HSF-1 turn-down in young adult life is blocked by *cbd-1* RNAi (and additional early eggshell/development gene RNAi) (*Sala et al., 2020*). Thus the presence of eggs can signal across tissues to turn-down *hsf-1* proteostasis-related activities in the mother. We speculate that this young adult reconfiguration of proteostasis might reflect a mechanism to optimize successful reproduction, possibly both fine-tuning nervous system function and shifting resources balance to favor progeny as suggested by the disposable soma theory of aging proposed by Kirkwood (*Kirkwood and Holliday, 1979*).

Of note, we do not observe exophers in larval stages (*Melentijevic et al., 2017*). We speculate that young adult physiology might be temporally tweaked such that some tissues have optimized capacity to manage/degrade large aggregates and organelles at an early adult developmental 'clean up' time, possibly analogous to how a town service for bulky oversized garbage pick-up might be limited to particular days during the year. As exopher production appears generally beneficial for neuronal function and survival (*Melentijevic et al., 2017*; *Yang et al., 2022*), the early life extrusion phase appears a positive feature of reproductive life. More broadly, proteome 'clean up' phases may be programmed as key steps at specific transitions during development and homeostasis, for example, as occurs in the temporal lysosome activation that clears aggregate debris in *C. elegans* maturing oocytes (*Bohnert and Kenyon, 2017*) or in the maturation of mouse adult neuronal stem cells via vimentin-dependent proteasome activity during quiescence exit (*Morrow et al., 2020*).

Across species, the production of exopher-like vesicles may also be enhanced by mechanical signals anchored outside of reproduction. For example, mice cardiomyocytes that are constantly under mechanical stress due to contraction activities produce exopher-like vesicles (*Nicolás-Ávila et al., 2020*). Mouse kidney proximal tubular epithelia cells (PTEC) under constant mechanical stress due to both fluid shear stresses and absorption-associated osmotic pressure, also release exopher-like vesicles (*Huang et al., 2023*).

## Large vesicle extrusion, mechanobiology, and neurodegenerative disease

The impact of mechanical force on in vivo production of extracellular vesicles has not been a major focus of the EV field, although a range of studies have considered force consequences (such as fluid shear responses, stretch) in cultured cells. Overall, however, EV biogenesis and uptake appear to be markedly influenced by biomechanical force type, magnitude, and duration (*Thompson and Papout-sakis, 2023*). At the same time, the neurodegeneration field has generated myriad studies linking Alzheimer's disease susceptibility and AD pathology signatures such as extracellular accumulation of amyloid-β protein and/or intracellular accumulation of tau as outcomes of mechanical stress-based stimuli such as traumatic brain injury, arterial hypertension, and normal pressure hydrocephalus (*Ramos-Cejudo et al., 2018*; *Malone et al., 2022*). Mechanical stress may trigger or promote protein misfolding, aggregation, and extrusion. Examples of recent implication of mechanical stimuli in AD-related outcomes include that stretch in the brain vascular system can increase APP and B-secre-tase expression to increase Aβ production (*Gangoda et al., 2018*) and that microglial mechano-sensing via the Piezo1 mechanotransducing channel limits progression of Aβ pathology in mouse models (*Hu et al., 2023*). Our study reveals a capacity of mechanical force to influence neuronal release of large vesicles containing neurotoxic species, inviting more serious consideration of the roles of mechanobiology in maintaining proteostasis and influencing aggregate transfer within the context of a living nervous system.

**Table 1.** Strain list.

| Strain Name | Genotype | Index |
|---|---|---|
| N2 | wild-type | wild-type |
| ZB4065 | bzIs166[P$_{mec-4}$::mCherry] II | wild-type |
| ZB4757 | bzIs166[P$_{mec-4}$::mCherry] II (ZB4065 outcrossed to N2 six times) | wild-type |
| ZB4768 | *glp-4(bn2)*$^{ts}$ I; bzIs166[P$_{mec-4}$::mCherry] II | *glp-4(ts)* |
| ZB5042 | bzIs166[P$_{mec-4}$::mcherry] II; *fem-3(q20)* IV | *fem-3(gf)* |
| ZB4915 | bzIs166[P$_{mec-4}$::mCherry] II; *fem-1(hc17)* IV | *fem-3(lf)* |
| ZB4749 | fxIs1[P$_{pie-1}$::TIR1::mRuby] zdIs5[P$_{mec-4}$::GFP] I; bzIs166[P$_{mec-4}$::mCherry] II; *spe-44(fx110[spe-44::degron])* IV. | SPE-44 |
| ZB4941 | bzIs166[P$_{mec-4}$::mCherry]; *gna-2(gk308)* I/hT2 [*bli-4(e937) let-?(q782)*] qIs48[P$_{myo-2}$::GFP; P$_{pes-10}$::GFP; P$_{ges-1}$::GFP] (I;III) | *gna-2(Δ)* |
| AD295 | *spe-45(tm3715)*; *him-5(e1490)*; asEx89 [*spe-45* 'fosmid 1' mixture +P$_{myo}$::gfp] | *spe-45(tm3715)* |
| ZB4772 | bzIs166[P$_{mec-4}$::mCherry] II; *egl-9(sa307)* V | *egl-9(Δ)* |
| ZB4904 | bzIs166[P$_{mec-4}$::mCherry] II; *egl-3(gk238)* V | *egl-3(Δ)* |
| ZB4902 | *sem-2(n1343)* I; bzIs166[P$_{mec-4}$::::mCherry] II | *sem-2(rf)* |
| ZB5352 | *goa-1(n1134)* I; bzIs166[P$_{mec-4}$::mCherry] II | *goa-1(Δ)* |
| ZB5046 | Ex [(pJC4) P$_{mec-3}$::gfp +pRF4]; bzIs166[P$_{mec-4}$::mCherry] II | pJC4 +*rol-6(su1006)* |
| ZB4942 | fxIs1[P$_{pie-1}$::TIR1::mRuby] I; bzIs166[P$_{mec-4}$::mCherry] II; *spe-44*(fx110[*spe-44*::degron]) IV; pwSi93[P$_{hyp7}$::oxGFP::*lgg-1*] | *Figure 5A* |
| ZB4953 | *sem-2(n1343)* fxIs1[P$_{pie-1}$::TIR1::mRuby] I; bzIs166[P$_{mec-4}$::mCherry] II; *spe-44(fx110[spe-44::degron])* IV | *sem-2(rf)* |
| ZB5709 | *sem-2(n1343)* I; bzIs166[P$_{mec-4}$::mCherry] II; *fem-3(q20)* IV. | *sem-2(rf); fem-3(q20)* |
| OD2984 | ltSi953 [P$_{mec-18}$::vhhGFP4::zif-1::operon-linker::mKate::*tbb-2* 3'UTR +*Cbr-unc-119(+)*] II; *unc-119(ed3)* III | Single-copy transgene |

## Materials and methods

### Strains and maintenance

All strains used in this study carry the transgene bzIs166[P*mec-4*::mCherry] to mark the six touch receptor neurons: ALMR, ALML, AVM, PVM, PLMR, and PLML. The genotype of *C. elegans* strains used in this study are listed in *Table 1*. We maintained all *C. elegans* strains on nematode growth media (NGM) seeded with OP50-1 *Escherichia coli* in a 20 °C or 15 °C incubator. We kept all animals on food for at least 10 generations before using them in a test.

### Age synchronization

For the majority of experiments, we used a bleaching protocol or an egg-laying protocol for age synchronization, otherwise, we picked L4 animals for synchronization.

### Temperature sensitive mutants

We maintained the age-synchronized temperature-sensitive mutations in a 15 °C incubator. For either *fem-1* or *fem-3* mutants, we directly placed the isolated eggs into a 25 °C incubator in each experimental test. Since the egg-hatching of *glp-4* mutant is out of sync at 25 °C, we placed the isolated eggs in a 15 °C incubator for 24 hr before transferring them into the 25 °C incubator for experimental tests.

### Auxin inducible degradation

We dissolved auxin (indole-3-acetic acid, 98+%, A10556, Alfa Aesar) in 95% ethanol to make a 400 mM stock, then we prepared a 40 mM auxin solution by diluting the 400 mM stock solution in M9 medium (which contains 1 mM MgSO4) and applied 200 ul of the 40 mM solution on to the NMG-agar plate (which is 60 mm in diameter and contains ~8–9 g medium). We left the plate on an open bench at room temperature for one or two days to dry out the auxin solution, then seeded the plate with 200 ul OP50-1 *E. coli* and waited for another two or three days before storing the plates in a 4 °C environmental room. We only used the plates which had been stored in the 4 °C environmental room for at least a week, so the concentration of auxin can be equilibrated into 1 mM. We prepared the ethanol control plates under the same procedure, and the final concentration of ethanol in the control plates should be ~0.25%.

To proceed auxin inducible degradation of SPE-44, we placed the isolated eggs on auxin-treated plates or the ethanol control plates. After ~72 hr of culture in a 20 °C incubator, the animal reached the age of adult day 1 and the auxin-treated worms became sterile. Then, we transferred the worms into a regular NGM-agar plate without auxin or ethanol.

### Microscopy and image processing

We took the DIC or fluorescent pictures with a Zeiss compound microscope or spinning disc confocal microscope driven by MetaMorph software, then processed the pictures with Fiji ImageJ software.

### Exopher scoring

Exopher numbers vary in experiments between 5–30%, mostly 10–15% range, and typically reach peak at adult day 2 (*Melentijevic et al., 2017*). Exphers can remain intact for approximately 2 hr, but the vesicle form of exophers is mostly identifiable in the following 24 hr. Therefore, the exopher count includes both the intact form and the fragmented degraded form, also known as 'starry night' (*Wang et al., 2023*). We scored the exopher count with the protocol published in JoVE (*Arnold et al., 2020*). We age-synchronized the animal via egg-laying, bleaching, or L4 picking. Then, we examined at least 50 animals for each genotype or treatment with the FBS10 Fluorescence Biological Microscope (KRAMER scientific), repeated for at least three trials.

### RNAi treatment

All RNAi clones used in this study come from the Ahringer RNAi library. The NGM-agar RNAi plate contains 1 mM IPTG and 25 µg/ml carbenicillin. The food on top of the medium is HT115 bacteria expressing dsRNA against a targeted gene or carrying an empty vector (EV, L4440) as the control. The treatment is from eggs to the last day of each test.

### FUdR treatment

The concentration of FUdR on NGM-agar plate is ~51 mM. The treatment started from adult day 1.

### Male mating experiment

In each trial, we prepared ~2000 age synchronized adult day 1 hermaphrodites and did exopher counting for 50 worms from adult day 1 to day 3. In adult day 3, we divided the worms into three groups (~400 worms in each group, and ~100 worms per plate). Group 1 is the control worms without males. We added ~100 sterile males into group 2 and ~100 normal males into group 3 for each plate.

### Electron microscopy

We prepared mCherry animals for TEM analysis by high pressure freezing and freeze substitution (HPF/FS), and followed a standard to preserve ultrastructure. After HPF in a Baltec HPM-010, we exposed animals to 1% osmium tetroxide, 0.1% uranyl acetate in acetone with 2% water added, held at −90 °C for 4 days before slowly warming back to −60 °C, −30 °C, and 0 °C, over a 2 day period. We rinsed the samples several times in cold acetone and embedded the samples into a plastic resin before curing them at high temperatures for 1–2 days. We collected serial thin sections on plastic-coated slot grids and post-stained them with 2% uranyl acetate and then with 1:10 Reynold's lead citrate, and examined with a JEOL JEM-1400 Plus electron microscope. By observing transverse sections for landmarks such as the 2nd bulb of the pharynx, it was possible to reach the vicinity of the ALM soma before collecting about 1500 serial thin transverse sections.

### The microinjection experiment

We mounted animals on coverslips with dried 2% agarose pads covered in halocarbon 700 oil. We then placed coverslips onto an Axiovert S100 TV Inverted Microscope (Carl Zeiss) and injected animals with capillary needles filled with M9 + 10% red dye under 40 psi of pressure. Needles were pulled from borosil capillary tubing (1.0 mm OD, 0.5 mm ID; FHC) using a P-97 Micropipette Puller (Sutter Instrument). Puller parameters were as follows: Heat 474 (Ramp-25); Pull 90; Vel 100; Time 180. For the control mock injection, we transiently poked the animal in the uterine region with a needle and then transferred the animal onto an NGM-agar plate. For the injected group, we applied injection pressure for 2 min with an estimated flow rate of approximately 25 fL/s and then transferred the animal onto an NGM-agar plate. After each injection or mock injection, we immediately examined the exopher status of the ALMR neuron in less than 10 min.

### Statistics

The exopher phenotype (+ or -) for each animal is a nominal variable, so we analyzed the exopher data by Cochran–Mantel–Haenszel test. There is no power analysis for each experiment, but each experiment has at least three independent trials. For analyzing progeny count, we used two-way ANOVA with Šídák's multiple comparisons test. For analyzing the uterus length, we used an unpaired two-tail *t*-test.

## Acknowledgements

We thank Dr. Andrew Singson, Dr. Amber Krauchunas, and Dr. Xue Mei for sharing reagents and providing comments in experimental design. We also thank the *Caenorhabditis* Genetics Center (CGC, founded by the National Institutes of Health - Office of Research Infrastructure Programs (P40 OD010440)) for providing some strains. Funding sources: NIH R24 OD010943 to DHH; NIH K12GM093854 to EC; NIH 5R01GM135326 to BDG; NIH R37AG56510 to MD; and NIH R01AG047101 to MD and BDG.

## Additional information

### Funding

| Funder | Grant reference number | Author |
|---|---|---|
| NIH Office of the Director | R24OD010943 | David H Hall |
| National Institute of General Medical Sciences | 5R01GM135326 | Barth D Grant |
| National Institute on Aging | R37AG56510 | Monica Driscoll |
| National Institute on Aging | R01AG047101 | Barth D Grant Monica Driscoll |
| National Institute of General Medical Sciences | K12GM093854 | Edward Chuang |

The funders had no role in study design, data collection and interpretation, or the decision to submit the work for publication.

### Author contributions

Guoqiang Wang, Ryan J Guasp, Sangeena Salam, Conceptualization, Data curation, Formal analysis, Validation, Investigation, Methodology, Writing – original draft, Project administration, Writing – review and editing; Edward Chuang, Conceptualization, Data curation, Validation, Investigation, Methodology, Writing – original draft; Andrés Morera, Anna J Smart, Conceptualization, Data curation, Formal analysis, Validation, Investigation, Methodology, Writing – original draft, Writing – review and editing; David Jimenez, Sahana Shekhar, Emily Friedman, Investigation; Ilija Melentijevic, Resources, Investigation; Ken C Nguyen, Validation, Investigation, Methodology; David H Hall, Resources, Data curation, Funding acquisition, Validation, Methodology, Writing – original draft; Barth D Grant, Conceptualization, Resources, Funding acquisition, Methodology; Monica Driscoll, Conceptualization, Resources, Supervision, Funding acquisition, Validation, Methodology, Writing – original draft, Project administration, Writing – review and editing

### Author ORCIDs

Guoqiang Wang ⬚ https://orcid.org/0000-0002-3694-7103
Barth D Grant ⬚ https://orcid.org/0000-0002-5943-8336
Monica Driscoll ⬚ https://orcid.org/0000-0002-8751-7429

Reviewer #1 (Public Review): https://doi.org/10.7554/eLife.95443.3.sa1
Reviewer #2 (Public Review): https://doi.org/10.7554/eLife.95443.3.sa2
Reviewer #3 (Public Review): https://doi.org/10.7554/eLife.95443.3.sa3
Author response https://doi.org/10.7554/eLife.95443.3.sa4

## Additional files

### Supplementary files
• MDAR checklist

### Data availability
Figure 1-source data, Figure 2-source data, Figure 3-source data, Figure 4-source data, Figure 5-source data, Figure 6-source data, and Figure 7-source data contain the numerical data used to generate the figures.

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
