## [Editor Report · eLife assessment]

This **important** study explores the potential influence of physiologically relevant mechanical forces on the extrusion of vesicles from *C. elegans* neurons. The authors provide **compelling** evidence to support the idea that uterine distension per se can induce vesicular extrusion from adjacent neurons. Overall, this work will be of interest to neuroscientists and investigators in the extracellular vesicle and proteostasis fields.

---

## [Referee Report · Reviewer #1 (Public Review)]

Summary:

The authors sought to understand the stage-dependent regulation of exophergenesis, a process thought to contribute to promoting neuronal proteostasis in *C. elegans*. Focusing on the ALMR neuron, they show that the frequency of exopher production correlates with the timing of reproduction. Using many genetic tools, they dissect the requirements of this pathway to eventually find that occupancy of the uterus acts as a signal to induce exophergenesis. Interestingly, the physical proximity of neurons to the egg zone correlates with exophergenesis frequency. The authors conclude that communication between the uterus and proximal neurons occurs through the sensing of mechanic forces of expansion normally provided by egg occupancy to coordinate exophergenesis with reproduction.

Strengths:

The genetic data presented is thorough and solid, and the observation is novel.

Weaknesses:

The authors have addressed the main weakness of the study in the revised manuscript, by providing data showing stimulation of exopher production in a single-copy transgenic line. Whether this process is related to the extrusion of cellular damage by the neurons in relatively young day 2 animals should be addressed in future studies.

---

## [Referee Report · Reviewer #2 (Public Review)]

Summary:

This paper reports that mechanical stress from egg accumulation is a biological stimulus that drives the formation of extruded vesicles from the neurons of *C. elegans* ALMR touch neurons. Using powerful genetic experiments only readily available in the *C. elegans* system, the authors manipulate oocyte production, fertilization, embryo accumulation, and egg-laying behavior, providing convincing evidence that exopher production is driven by stretch-dependent feedback of fertilized, intact eggs in the adult uterus. Shifting the timing of egg production and egg laying alters the onset of observed exophers. Pharmacological manipulation of egg laying has the predicted effects, with animals retaining fewer eggs having fewer exophers and animals with increased egg accumulation having more. The authors show that egg production and accumulation have dramatic consequences to the viscera, and moving the ALMR process away from eggs prevents the formation of exophers. This effect is not unique to ALMR but is also observed in other touch neurons, with a clear bias toward neurons whose cell bodies are adjacent to the filled uterus. Embryos lacking an intact eggshell with reduced rigidity have impaired exopher production. Acute injection into the uterus to mimic the stretch that accompanies egg production causes a similar induction of exopher release. Together these results are consistent with a model where stretch caused by fertilized embryo accumulation, and not chemical signals from the eggs themselves or egg release, underlies ALMR exopher production seen in adult animals.

Strengths:

Overall, the experiments are very convincing, using a battery of RNAi and mutant approaches to distinguish direct from indirect effects. Indeed, these experiments provide a model generally for how one would methodically test different models for exopher production. The source and factors influencing exopher production had previously been unclear. This study addresses how and when they form in the animal using physiologically meaningful manipulations. The stage is now set to address at a cellular level how exophers like these are made and what their functions are.

Weaknesses:

Not many. The experiments are about as good as could be done. Some of the n's on the more difficult to work strains or experiments are comparatively low, but this is not a significant concern because the number of different, complementary approaches used. The microinjection experiment is very interesting, and the authors have added additional details on how these experiments were conducted in the revised manuscript. The authors have now included data from strains bearing a single-copy transgene that expresses mKate2 in the same neurons, showing that induced egg accumulation drives a similar degree of exopher production. This indicates that exposers seen are generated in response to specific biological conditions and not merely an artifact of mCherry protein over-expression.

---

## [Referee Report · Reviewer #3 (Public Review)]

Summary:

In this paper, the authors use the *C. elegans* system to explore how already-stressed neurons respond to additional mechanical stress. Exophers are large extracellular vesicles secreted by cells, which can contain protein aggregates and organelles. These can be a way of getting rid of cellular debris, but as they are endocytosed by other cells can also pass protein, lipid, and RNA to recipient cells. The authors find that when the uterus fills with eggs or otherwise expands, a nearby neuron (ALMR) is far more likely to secrete exophers. This paper highlights the importance of the mechanical environment in the behavior of neurons and may be relevant to the response of neurons exposed to traumatic injury.

Strengths:

The paper has a logical flow and a compelling narrative supported by crisp and clear figures.

The evidence that egg accumulation leads to exopher production is strong. The authors use a variety of genetic and pharmacological methods to show that increasing pressure leads to more exopher production, and reducing pressure leads to lower exopher production. For example, egg-laying defective animals, which retain eggs in the uterus, produce many more exophers, and hyperactive egg-laying is accompanied by low exopher production. The authors even inject fluid into the uterus and observe the production of exophers.

Weaknesses:

The main weakness of the paper is that it does not explore the molecular mechanism by which the mechanical signals are received or responded to by the neuron. The authors are currently addressing this in their follow-up studies.

---

## [Author Response]

The following is the authors’ response to the original reviews.

**eLife assessment**
This important study explores the potential influence of physiologically relevant mechanical forces on the extrusion of vesicles from *C. elegans* neurons. The authors provide compelling evidence to support the idea that uterine distension can induce vesicular extrusion from adjacent neurons. The work would be strengthened by using an additional construct (preferably single-copy) to demonstrate that the observed phenotypes are not unique to a single transgenic reporter. Overall, this work will be of interest to neuroscientists and investigators in the extracellular vesicle and proteostasis fields.

We now include supporting data using a single copy alternate fluorescent reporter expressed in touch neurons (Fig. 3H).

In brief, we examined the induction of exophergenesis in an alternative single-copy transgene strain that expresses mKate fluorescent protein specifically in touch receptor neurons. As compared to the multi-copy transgene that is broadly used in this study and expresses mCherry fluorescent protein specifically in touch receptor neurons, the mKate single-copy transgene is associated with a much lower frequency of exophergenesis. However, increasing uterine distension via blocking egg-laying can increase the exophergenesis of the mKate single-copy transgenic line from 0% to approximately 60% on adult day 1, indicating that the observed response is not tied to a single reporter.

**Public Reviews:**

**Reviewer #1 (Public Review):**
Summary:The authors sought to understand the stage-dependent regulation of exophergenesis, a process thought to contribute to promoting neuronal proteostasis in *C. elegans*. Focusing on the ALMR neuron, they show that the frequency of exopher production correlates with the timing of reproduction. Using many genetic tools, they dissect the requirements of this pathway to eventually find that occupancy of the uterus acts as a signal to induce exophergenesis. Interestingly, the physical proximity of neurons to the egg zone correlates with exophergenesis frequency. The authors conclude that communication between the uterus and proximal neurons occurs through the sensing of mechanic forces of expansion normally provided by egg occupancy to coordinate exophergenesis with reproduction.Strengths:The genetic data presented is thorough and solid, and the observation is novel.Weaknesses:The main weakness of the study is that the detection of exophers is based on the overexpression of a fluorescent protein in touch neurons, and it is not clear whether this process is actually stimulated in wild-type animals, or if neurons have accumulated damaged proteins in relatively young day 2 animals.

We now include data using a single copy alternate fluorescent reporter expressed in touch neurons. Although baseline exopher levels are low in this strain, we demonstrate that inducing egg retention in this background markedly increases exopher generation from a baseline of near zero to ~60% (new Fig. 3H), supporting that uterine distention, rather than reporter identity, is associated with early life exopher elevation. Data also add to our observations indicating that high protein-expressing strains generally produce higher baseline levels of exophers in early adulthood (for example, Melentijevic et al. (PMID 28178240) documented that mCherry RNAi knockdown in the strain primarily studied here can lower exopher levels).

The second point raised here, regarding the occurrence and physiological role of early-adult exophers in “native” non-stressed neurons is a fascinating question that we are beginning to address in continuing experiments. Readers will appreciate that quantifying relatively rare, “invisible” touch receptor neuron exophergenesis accurately without expressing a fluorescent reporter is technically challenging. Our speculation, outlined now a bit more clearly in the Discussion here, is that certain molecular and organelle debris that cannot readily be degraded in cells during larval development may be stored until release to more capable degradative neighbors or to the coelomocytes for later management, as one component of the early adult transition in proteostasis (see J. Labbadia and R. I. Morimoto, PMID 24592319). Receiving cells may be primed for this at a particular timepoint, possibly analogous to the “bulky garbage” collection of over-sized difficult-to-dispose-of household items that a town will address with specialized action only at specific times. The prediction is that we should be able to detect some mass protein aggregation through early development, and at least partial elimination by adult day 3; this elimination should be impaired when eggs are eliminated. Initial testing is underway.

**Reviewer #2 (Public Review):**
Summary:This paper reports that mechanical stress from egg accumulation is a biological stimulus that drives the formation of extruded vesicles from the neurons of *C. elegans* ALMR touch neurons. Using powerful genetic experiments only readily available in the *C. elegans* system, the authors manipulate oocyte production, fertilization, embryo accumulation, and egg-laying behavior, providing convincing evidence that exopher production is driven by stretch-dependent feedback of fertilized, intact eggs in the adult uterus. Shifting the timing of egg production and egg laying alters the onset of observed exophers. Pharmacological manipulation of egg laying has the predicted effects, with animals retaining fewer eggs having fewer exophers and animals with increased egg accumulation having more. The authors show that egg production and accumulation have dramatic consequences for the viscera, and moving the ALMR process away from eggs prevents the formation of exophers. This effect is not unique to ALMR but is also observed in other touch neurons, with a clear bias toward neurons whose cell bodies are adjacent to the filled uterus. Embryos lacking an intact eggshell with reduced rigidity have impaired exopher production. Acute injection into the uterus to mimic the stretch that accompanies egg production causes a similar induction of exopher release. Together these results are consistent with a model where stretch caused by fertilized embryo accumulation, and not chemical signals from the eggs themselves or egg release, underlies ALMR exopher production seen in adult animals.Strengths:Overall, the experiments are very convincing, using a battery of RNAi and mutant approaches to distinguish direct from indirect effects. Indeed, these experiments provide a model generally for how one would methodically test different models for exopher production. The paper is well-written and easy to understand. I had been skeptical of the origin and purpose of exophers, concerned they were an artefact of imaging conditions, caused by deranged calcium activity under stressful conditions, or as evidence for impaired animal health overall. As this study addresses how and when they form in the animal using otherwise physiologically meaningful manipulations, the stage is now set to address at a cellular level how exophers like these are made and what their functions are.Weaknesses:Not many. The experiments are about as good as could be done. Some of the n's on the more difficult-to-work strains or experiments are comparatively low, but this is not a significant concern because of the number of different, complementary approaches used. The microinjection experiment in Figure 7 is very interesting, there are missing details that would confirm whether this is a sound experiment.

We expanded description of details for the microinjection experiment in both the figure legend and the methods section, to enhance clarity and substantiate approach.

**Reviewer #3 (Public Review):**
Summary:In this paper, the authors use the *C. elegans* system to explore how already-stressed neurons respond to additional mechanical stress. Exophers are large extracellular vesicles secreted by cells, which can contain protein aggregates and organelles. These can be a way of getting rid of cellular debris, but as they are endocytosed by other cells can also pass protein, lipid, and RNA to recipient cells. The authors find that when the uterus fills with eggs or otherwise expands, a nearby neuron (ALMR) is far more likely to secrete exophers. This paper highlights the importance of the mechanical environment in the behavior of neurons and may be relevant to the response of neurons exposed to traumatic injury.Strengths:The paper has a logical flow and a compelling narrative supported by crisp and clear figures.The evidence that egg accumulation leads to exopher production is strong. The authors use a variety of genetic and pharmacological methods to show that increasing pressure leads to more exopher production, and reducing pressure leads to lower exopher production. For example, egg-laying defective animals, which retain eggs in the uterus, produce many more exophers, and hyperactive egg-laying is accompanied by low exopher production. The authors even inject fluid into the uterus and observe the production of exophers.Weaknesses:The main weakness of the paper is that it does not explore the molecular mechanism by which the mechanical signals are received or responded to by the neuron, but this could easily be the subject of a follow-up study.

We agree that the molecular mechanisms operative are of considerable interest, and our initial pursuit suggests that a comprehensive study will be required for satisfactory elaboration of how mechanical signals are received or responded to by the neuron.

I was intrigued by this paper, and have many questions. I list a few below, which could be addressed in this paper or which could be the subject of follow-up studies.- Why do such a low percentage of ALMR neurons produce exophers (5-20%)? Does it have to do with the variability of the proteostress?

We do not yet understand why some ALMR neurons within a same genotype will produce exophers and some will not. We know that in addition to the uterine occupation we report here, proteostasis compromise, feeding status, oxidative stress, and osmotic stress can elevate exopher numbers (PMID 34475208); cell autonomous influences on exopher levels include aggresome-associated biology (PMID 37488107) and expression levels of the mCherry protein (PMID 28178240). Turek reports that social interaction on plates can influence muscle exopher levels (PMID 34288362). Thus, although variable proteostress experienced by neurons is likely a factor, we have not yet experimentally defined specific trigger rules. We suspect the summation of internal proteostasis crisis and environmental conditions, including particular force vectors/frequency will underlie the variable exopher production phenomeonon.

- Why does the production of exophers lag the peak in progeny production by 24-48 hours? Especially when the injection method produces exophers right away?

The progeny production can track well with exopher production (Fig. 1B), although the nature of egg counts (permanent, one time events) vs. exophers (which are slowly degraded) can skew the peak scores apart. We synchronized animals at the L4 stage. 24 hours later was adult day 1, and we measured then and every subsequent 24 hours. The daily progeny count reflects the total number of progeny produced every 24 hours; exopher events were scored once a day, but exophers can persist such that the daily exopher count can partially reflect slow degradation, with some exophers being counted on two days. We now explain our scoring details better in the Methods section.

The rapid appearance of exophers, as early as about ~10 minutes after sustained injection, is fascinating and probably holds mechanistic implications for exopher biology. For one thing, we can infer that in the mCherry Ag2 background, touch neurons can be poised to extrude exophers, but that the pressure/push acts to trigger or license final expulsion. It is interesting that we found we needed to administer sustained injection of two minutes to find exopher increase (now better emphasized in the expanded Methods section). We speculate that a multiple pressure events, or sustained force vector might be critical (like an egg slowly passing through??). Minimally, this assay may help us assign molecular roles to pathway components as we identify them moving forward.

- As mentioned in the discussion, it would be interesting to know if PEZO-1/PIEZO is required for uterine stretching to activate exophergenesis. pezo-1 animals accumulate crushed oocytes in the uterus.

We have begun to test the hypothesis that PEZO-1 is a signaling component for ALMR exophergenesis, initially using the N and C terminal pezo-1 deletion mutants as in Bai et al. (PMID 32490809). These pezo-1 mutants have a mild decrease in ALMR exophergenesis under normal conditions. However, vulva-less conditions in pezo-1N and piezo-1C increased ALMR exophergenesis from approximately 10% to 60%, similar to the response of wild-type worms to high mechanical stress, data that suggest PEZO-1 is not a required player in mediating mechanical force-induced ALMR exophergenesis. We are currently testing genetic requirements for other known mechanosensors. We intend comprehensive investigation of the molecular mechanisms of mechanical signaing in a future study.

**Recommendations for the authors:**

**Reviewer #1 (Recommendations For The Authors):**
-The study would be significantly strengthened by the addition of data detecting regulation of exophergenesis by uterine forces in a more physiological context, in the absence of overexpression of a toxic protein. In other words, is this a process that occurs naturally during reproduction, or is it specific to proteotoxic stress induced by overexpression? Perhaps the authors could repeat key experiments using a single copy transgene, and challenge the animals with exogenous proteotoxic stress if necessary.

We now include data using a single copy alternate fluorescent reporter expressed in touch neurons. Although baseline exopher levels are low in this strain, we demonstrate that inducing egg retention in this background markedly increases exopher generation from a baseline of near zero to ~60% (Fig. 3H), supporting that uterine distention, rather than reporter identity or over-expression alone dries early life exopher elevation.

Also noteworthy is that we find exophergenesis in the single-copy transgenic line is only approximately 0.3% on adult day 2 (average in three trials, data not shown), which is much lower than the 5-20% exophergenesis rate typically observed in the multi-copy high expression mCherry transgenic line. Therefore, consequences of overexpression of mCherry likely potentiate exophergenesis.

-The authors mention that exophergenesis has been described in muscle cells. Is this also dependent on the proximity to the uterus? It would have been interesting to include data on other cell types in the vicinity of the reproductive system.

Yes, in interesting work on exophers produced by muscle, Turek et al. reported that muscle exopher events are mostly located in a region proximal to the uterus. Moreover, this work also documented that sterile hermaphrodites are associated with approximately 0% muscle exophergenesis, and egg retention in the uterus strongly increases muscle exophergenesis (PMID: 34288362).

-Is exophergenesis also induced by other forms of mechanical stress? For example, swimming.

We have looked at crude treatments such as centrifugation or vortexing without observing changes in exopher levels. Our preliminary work indicates that swimming can increase exophergenesis, and this effect depends on the presence of eggs in the uterus. We appreciate the question, and expect to include documentation of alternative pressure screening in our planned future paper on molecular mechanisms.

-In Figure 1E, the profile of exopher production for the control condition at 25oC is very similar to the profile observed at 20oC in Figure 1B. However, the profile of progeny production at 25oC is known to have an earlier peak of progeny production. Perhaps egg retention is differently correlated with progeny production at this temperature? The authors could easily test this.

Overall, exophers (which degrade with time) and progeny counts (a fixed number) have slightly different temporal features, anchored in part by how long exophers or their “starry night” debris persist. Most exophers start to degrade within 1-6 hours (PMID: 36861960), but exopher debris can persist for more than 24 hours. An exopher event observed on day 1 may thus also be recorded at the day 2 time point, which leads to a higher frequency of exopher events on day 2 as compared to day 1.

We have previously published on the impact of temperature on exopher number (Supplemental Figure 2 in PMID 34475208). In brief, increasing culture temperature for animals that are raised over constant lifetime temperature modestly increases exopher number; a greater increase in exophers is observed under conditions in which animals were switched to a higher temperature in adult life, suggesting changes in temperature (a mandatory part of the ts mutant studies) engages complex biology that modulates exopher production. Our previous data show that in a temperature shift to 25oC, the peak of exophers was at adult day 1. Here, Fig. 1B is constant temperature, 20oC; Fig. 1E has a temperature shift 15-25oC. That egg retention might be temperature-influenced is a plausible hypothesis, but given the complexities of temperature shifts for some mutants, we elected to defer drill-down on the temperature-exopher-egg relationship.

-It is not clear how to compare panels A and B in Figure 3. In panel A the males are present throughout the adult life of the hermaphrodites whereas in panel B the males are added in later life. Therefore, the effect of later-life mating on progeny production is not shown and the title of panel A in the legend is misleading. The authors need to perform a progeny count in the same conditions of mating presented in Figure 3B to allow direct comparison.

As Reviewer 1 suggested, we performed a new progeny count now presented in new Fig. 3A, which more appropriately matches the study presented in Fig. 3B; legends adjusted.

-On page 12, the authors state that the baseline of exophergenesis in rollers is 71%, but then attribute the 71% in Figure 4F to exophergenesis specifically in ALMR that is posterior to AVM. The authors need to clarify this point.

Good catch on our error. The baseline of exophergenesis in rollers is ~40%, and we corrected the main text.

-Considering the conclusion of Figure 2 that blocking embryonic events passed the 4-cell stage does not impact exopher production, it would have been interesting to compare the uterine length for emb-8 and for mex-3, since it is quite intriguing that the former suppresses exopher production while the latter has no effect.

We repeated the emb-8 and mex-3 RNAi for these studies and encountered variability in outcome for 2 cell stage disruption via emb-8 RNAi, which is consistent with the range of published endpoints for emb-8 RNAi. We elected to include these emb-8 findings in the figure legend 2G, but removed the RNAi data from the main text figure. mex-3 uterine measures are added to revised panels 5H, 6I.

**Reviewer #2 (Recommendations For The Authors):**
-Leaving the worms in halocarbon oil for too long (e.g. 10 min) can desiccate and kill them. Did the authors take them out of the oil before analyzing exopher production? The authors refer to these as 'sustained injections' without much description beyond that. As the worms are very small, the flow rate needed for a sustained injection over 2 minutes must be very low - so low that the needle is in danger of being clogged. Do the authors have an estimate of how much fluid was injected or the overall flow rate? I realize the flow rate measured outside of the worm may not compare directly to that of a pressurized worm, but such estimates would be instructive, particularly if they can be related to the relative volume of the eggs the injection is trying to mimic.

After injection or mock injection, we removed the animal from the oil and flipped it if necessary to observe the ALMR neuron on the NGM-agar plate. We now expanded description of the experimental details of injection, including the estimated flow rate, in the revised Methods section.

- The authors describe the ALMR neurons as "proteostressed", but I am not clear on whether these neurons were treated in a unique procedure to induce such a state or if the authors are merely building on other observations that egg-laying adults are dedicating significant resources to egg production, so they must be proteostressed. If they are not inducing a proteostressed state in their experiments, the authors should refrain from describing their neurons and effects as depending on such a state.

We revised to more explicity feature published evidence that the ALMR neurons we track with mCherryAg2 bz166 are likely protestressed. Overexpression of mCherry in bz166 is associated with enlargement of lysosomes and formation of large mCherry foci that often correspond toe LAMP::GFP-positive structures in ALMR neurons (PMID: 28178240; PMID: 37488107). Marked changes in ultrastructure reflect TN stress in this background. These cellular features are not seen in wild type animals. We previously published that mCherry, polyQ74, polyQ128, Ab1-42 (which enhance proteostress) over-expression all increase exophers (PMID: 28178240). Likewise most genetic compromise of different proteostasis branches--heat shock chaperones, proteasome and autophagy--promote exophergenesis, supporting exophergenesis as a response to proteostress. In sum, the mCherryAg2 bz166 appear markedly stressed above a non-over expressing line and produce more exophers. RNAi knockdown of the mCherry lowers exopher levels (PMID: 28178240).

In response to reviewer comment, we added a study with a single copy mKate reporter (new data Fig. 3H). We find a very low baseline of exophers in this background. This would support that high autonomous compromise associated with over-expression influences exopher levels. Interestingly, however, we found that ALMR neurons expressing mKate under a single-copy transgene still exhibit excessive exopher production (>60%) under high mechanical stress (Fig. 3H). These data are consistent with ideas that mechanical stresses can enhance exopher production, and may markedly lower the threshold for exophergenesis in close-to-native stress level neurons.

- The authors should include more details on the source and use of the RNAi, for example, if the clones were from the Ahringer RNAi library, made anew for this study, or both.

We now add this information in the methods section.

- I would be curious if the authors would similarly see an induction in exopher production after acute vulval muscle silencing with histamine. I'm not suggesting this experiment, but it may offer a way to induce exophers in a more controlled manner.

This is a great suggestion that we will try in future studies.

- I am not sure if Figure 5 needs to be a main figure in the paper or if it would be more appropriate as a supplement.

We considered this suggestion but we think that the strikingly strong correleation of uterus length and exopher levels is a major point of the story and these data establish a metric that we will use moving forward to distinquish whethere an exopher modulation disruption is more likely to act by modulation of reproduction or modulation of touch neuron biology. For this reason we elected to keep Figure 5 in the main text.

**Reviewer #3 (Recommendations For The Authors):**
-The Statistics section in the methods should be expanded to describe the statistics used in the experiments that aren't nominal, of which there are many.

We have updated and expanded the statistics section.

-P.2 Line 49 spelling 'que' should be queue (I remember this by the useless queue of letters lined up after the 'q').

Corrected

-The introduction has a bit too much information about oocyte maturation, not relevant to the study.

We agree that the information about oocyte maturation is not critical for the laying out the related experiments and cut this section to improve focus.

-p.3 line 22: Some exophers are seen on Day 3, so this should be restated for accuracy.

Corrected

-p.3 line 26. Explain here why sperm is necessary (ooyctes don't mature or ovulate effectively without sperm).

We added this clarifying explanation.

-p.3 line 44 Clarify in the spe-44 the oocytes are in the oviduct (not the uterus). Might be helpful to include a DIC image to accompany the helpful diagram in Figure 1D.

We added a sentence describing the impact of sperm absence on oocyte maturation, progression into the uterus, and retention in the gonad, with reference to PMID: 17472754. We were able to add a DIC in the tightly packed Figure 1.

In Supplemental Figure 6, we now include a field picture of oocyte retention in the sem-2 mutant and upon treatment of lin-39(RNAi).

-p.5 line 3 in the Figure 1D legend; recommend delete 'light with' which is confusing and just refer to the sperm as dark dots.

Corrected

-p.6 line 22-24 Check for alignment of the statements with Figure 2 (2F is cited, but it should be 2G).

Corrected

-p12 line 13-15; Many ALMRs not in the egg zone (70%) did not produce exophers - this is still quite a lot. It would be good to state this section in a more straightforward way (less leading the reader) and if possible to give a possible explanation.

We modified the text to be less leading: “Thus, although ALMR soma positioning in the egg zone does not guarantee exophergenesis in the mCherryAg2 strain, the neurons that did make exophers were nearly always in the egg zone.”

-p.15 paragraph 3 - clarify how uterine length was controlled for the overall body length of the worm.

We did not systematically measure body length, but rather focused on uterine distention. It would be of interest to determine if length of the body correlates with uterine size, and then address how that relationship translates to exopher production but here our attention came to rest on the striking correlation of uterine length and number of exophers.

-p.17 line 23-25; Could be stated more simply.

We adjusted the text: “Moreover, the oocyte retention was similarly efficacious in elevating exopher production to egg retention, increasing ALMR exophergenesis to approximately 80% in the sem-2(rf) mutant (Fig. 6C)”.

-p.23 Line 4. I think by the time the reader reaches this sentence, the egg-coincident exophorgenesis will not be 'puzzling'.

Agreed, corrected.

-p.26, Line 22, Male 'mating', not 'matting'.

Corrected.

-Throughout, leave space between number and unit (this is not required for degree or percent, but be consistent).

Corrected.